

1    **Remote-sensing estimate of glacier mass balance over the central**

2    **Nyainqentanglha Range during 1968 – ~2013**

Kunpeng Wu [1, 2*], Shiyin Liu [2, 3*], Zongli Jiang [4], Junli Xu [5], Junfeng Wei [4]

[1]School of Resources and Environment, Anqing Normal University, Anqing, 246133, China

[2]Institute of International Rivers and Eco-Security, Yunnan University, Kunming, 650091, China

[3]State Key Laboratory of Cryospheric Sciences, Northwest Institute of Eco-Environment and Resources, Chinese Academy of

Sciences, Lanzhou, 730000, China

[4]Department of Geography, Hunan University of Science and Technology, Xiangtan, 411201, China

[5]Department of Surveying and Mapping, Yancheng Teachers University, Yancheng, 224007, China

Correspondence to LIU Shiyin at liusy@lzb.ac.cn or WU Kunpeng at wukunpeng2008@lzb.ac.cn

* These authors contributed equally to this work and should be considered co-first authors

**Abstract.** With high air temperatures and annual precipitation, maritime glaciers in southeastern Tibet

are sensitive to climate change. Current glaciological knowledge of those in the central

Nyainqentanglha Range is still limited because of their inaccessibility and low-quality data. To obtain

information on changes in glacier area, length and mass balance, a comprehensive study was carried

out based on topographic maps and Landsat TM/ETM+/OLI images (1968 and 2016), and on

digital-elevation models (DEM) derived from the 1968 maps, from the Shuttle Radar Topography

Mission (SRTM) DEM (2000), and from TerraSAR-X/TanDEM-X (~2013). This showed the area

contained 715 glaciers, with an area of 1713.42 $\pm$ 51.82 km$^2$, in 2016. Ice cover has been shrinking by

0.68% $\pm$ 0.05% a$^{-1}$ since 1968, although in the most recent decade this rate has slowed. The glacier area

covered by debris accounted for 11.9% of the total and decreased in SE-NW directions. Using DEM

differencing and Differential Synthetic Aperture Radar Interferometry (DInSAR), a significant mass

deficit of 0.46 $\pm$ 0.04 m w.e. a$^{-1}$ has been recorded since 1968; mass losses accelerating from 0.42 $\pm$

0.05 m w.e. a$^{-1}$ to 0.60 $\pm$ 0.20 m w.e. a$^{-1}$ during 1968–2000 and 2000–~2013, with thinning noticeably

greater on the debris-covered ice than the clean ice. Surface-elevation changes can be influenced by ice

cliffs, as well as debris cover, and land- or lake-terminating glaciers and supraglacial lakes. Changes

showed spatial and temporal heterogeneity and a substantial correlation with climate warming.

**1    Introduction**

The Tibetan Plateau (TP), known as the roof of the world or Third Pole, contains the largest

concentration of glaciers and icefield outside the Polar Regions (Yao, 2008). Meltwater from these

feeds the headwaters of many prominent Asian rivers (e.g., the Yellow, Yangtze, Mekong, Salween,

Brahmaputra, Ganges and Indus) (Immerzeel et al., 2010), and are a key component of the cryospheric

system (Li et al., 2008). Glaciers are important climate indicators because their extent and thickness

adjust in response to climate change (Oerlemans, 1994; T. Yao et al., 2012). With a warming climate,

many mountain glaciers have shrunk progressively in mass and extent during past decades (IPCC,

2013). However, slight mass gains or balanced mass budgets have been evident for parts of the central

Karakoram, eastern Pamir and the western TP in recent years (Bao et al., 2015; Gardelle et al., 2012b;





Gardelle et al., 2013; Kääb et al., 2015; Ke et al., 2015; Neckel et al., 2014; T. Yao et al., 2012). The
relationships between glacier mass balance and climate change, water supply and the risk of
glacier-related disasters, are the subject of much current research.

It is difficult to carry out in-situ observations on the Tibetan Plateau due to its rugged terrain and
the great labor and logistical costs. Only 15 glaciers have decades of mass-balance measurements (T.
Yao et al., 2012). Fortunately, new methods are now available for estimating large-scale glacier mass
balance, such as satellite geodesy. By comparing topographic data from more than two points in time,
glacier volume or height changes can be determined and thence glacier mass balance, after
consideration of ice/firn/snow densities (Bolch et al., 2011; Gardelle et al., 2013; Kääb et al., 2012;
Paul et al., 2015; Pieczonka et al., 2013; Shangguan et al., 2014).

Glaciers in south-eastern Tibet are reportedly of the temperate (maritime) type and are influenced
by the South Asian monsoon (Li et al., 1986; Shi and Liu, 2000). Based on inventories from maps and
remote sensing, or field measurements, a substantial reduction in glacier area and length has been
recorded from 1980–2013, as well as a glacier mass deficit from 2005–2009 (Li et al., 2014; Yang et
al., 2010; Yang et al., 2008; T. Yao et al., 2012). Most previous studies used satellite laser or optical
photogrammetry to calculate the glacier height changes that determined pronounced negative glacier
mass balances in the region (Gardelle et al., 2013; Gardner et al., 2013; Kääb et al., 2015; Neckel et al.,
2014), although the results did differ slightly from each other. ICESat footprint data showed geodetic
glacier-elevation difference trends in south-eastern Tibet during 2003–2008 of $1.34 \pm 0.29$ m a$^{-1}$ (Kääb
et al., 2015), $0.81 \pm 0.32$ m a$^{-1}$ (Neckel et al., 2014) and $0.30 \pm 0.13$ m a$^{-1}$ (Gardner et al., 2013),
respectively. The large orbital gaps in these data mean spatial details cannot be mapped at a fine scale,
whereas photogrammetry does provide better spatial detail on glacier-height changes. A comparison
between SPOT5/HRS and Shuttle Radar Topography Mission (SRTM) data from January 2011 found a
mean glacier thinning of $0.39 \pm 0.16$ m a$^{-1}$ in south-eastern Tibet (Gardelle et al., 2013). However, the
lack of local mass-balance measurements means details on the specific response of these glaciers to
climate change were lacking, especially for the western region which is the central Nyainqentanglha
Range (CNR). Bistatic SAR interferometry is an alternative method to optical photogrammetry and
altimetry for analysing topographic change. TanDEM-X was launched in 2010 to join its twin satellite,
TerraSAR-X, and operates with it in bistatic mode. This mode overcomes the temporal decorrelation
and atmospheric-delay disturbance associated with conventional repeat-pass interferometry (Jaber et al.,
2012). Based on the Shuttle Radar Topography Mission (SRTM) and an interferometrically derived
TanDEM-X elevation model, glaciers were determined to have experienced strong surface lowering in
the CNR, at an average rate of $-0.83 \pm 0.57$ m a$^{-1}$ from 2000–2014 (Neckel et al., 2017). While this
pronounced surface-lowering value came from five debris-covered valley glaciers in the study area, it
cannot represent large-scale glaciers' response to climate warming. Furthermore, most meteorological
stations, located in inhabited river valleys, are far from the glacierized high mountain regions so their
records cannot be used directly as climate background for them. Even when in the same climate
environment, glaciers are also responding to local parameters, such as catchment aspect, topography,
and debris cover (Kääb, 2005; Neckel et al., 2017; Scherler et al., 2011).

Topographic Maps were drawn from aerial photographs taken in April 1968, and subsequently the
Shuttle Radar Topography Mission (SRTM) DEM resulted by X-band SAR Interferometry (InSAR) in
February 2000. Single-pass X-band InSAR from TerraSAR-X and TanDEM-X digital elevation
measurements provided the basis for another map (Krieger et al., 2007). Bistatic Differential Synthetic
Aperture Radar Interferometry (DInSAR) and common DEM differencing were used to estimate the



1    geodetic glacier mass balance in different sub-regions of the CNR between 1968 and ~2013.

## 2    Study area

The CNR lies in south-eastern Tibet, north of Linzhi County, east of Jiali County and west of
Bomi County, extending about 130 km from west to east. South of this region is the Yigong Tsangpo
River, a tributary of the Purlung Tsangpo River and a secondary tributary of the Yarlung Tsangpo River
(Fig. 1). The southern slopes are exposed to the moist southwest monsoon (Li et al., 1986) which enters
the plateau at the Grand Bend of the Yarlung Zangbo. Because the terrain forces the air to rise, the
CNR is the most humid region of the Tibetan Plateau and one of the most important and concentrated
regions of maritime (temperate) glacier development (Shi et al., 2008; Shi and Liu, 2000). The mean
summer air temperature at the equilibrium-line altitude (ELA) of glaciers here is usually above 1 °C,
and annual precipitation is 2500–3000 mm (Shi et al., 1988).

The first Chinese Glacier Inventory (CGI) determined that glaciers covered 2537.7 km$^2$ of our
study region, with a total volume of 454.2 km$^3$ (Mi et al., 2002; Pu, 2001); about 8% of the area was
covered by debris. Three glaciers in the CNR are larger than 100 km$^2$, the Xiaqu (CGI code:
5O281B0702), Kyagqen (CGI code: 5O281B0729) and Nalong (CGI code: 5O281B0768). The
Kyagqen, on the south slope of the CNR, 35.3 km long and 206.7 km$^2$, with a terminus at 2900 m a.s.l.,
is the largest of these (Li et al., 1986). Above 4000 m a.s.l. it has a broad basin in which several ice
streams converge to form a large accumulation zone (165 km$^2$) that accounts for over 80% of the
glacier's total area. Below this, the glacier enters a narrow ice-filled valley where its velocity increases;
the resultant great driving force pushing the glacier terminus to a subtropical elevation at 2900 m a.s.l.
The narrow glacier tongue, 1000 m wide and 17 km long, passes through the subalpine shrub-meadow
zone, the mountain dark coniferous forest zone and the mixed broadleaf-conifer forest zone (Li et al.,
1986).

## 3    Data

### 3.1    Topographic Maps

Our study uses eight topographic maps at a scale of 1:100 000 (Fig. 1 and Table 1). They were
compiled by the Chinese Military Geodetic Service from air photos acquired in April 1968. Their
geographic projection was based on the Beijing Geodetic Coordinate System 1954 (BJ54) geoid and
the Yellow Sea 1956 datum. Using a seven parameter transformation method, these maps were
re-projected into the World Geodetic System 1984 (WGS1984)/Earth Gravity Model 1996 (EGM96)
(Xu et al., 2013). The contour lines were digitized from topographic maps manually, and then using the
Thiessen polygon method, converted into a raster DEM with a 30 m grid cell (hereafter called TOPO
DEM) (Shangguan et al., 2010; Wei et al., 2015; Zhang et al., 2016a). According to the national
photogrammetric standard of China, the vertical accuracy of the TOPO DEM is better than 16 m on
glaciers with gentle slopes (~24 °) which is common for most of the glacierized areas in the CNR.

### 3.2    Shuttle Radar Topography Mission

Acquired by radar interferometry with C-band and X-band in early February 2000, the SRTM DEM
can be referred to the glacier surface in 1999 with slight seasonal variances (Gardelle et al., 2013;
Pieczonka et al., 2013; Zwally et al., 2011). Due to large data gaps in the X-band DEM (Rabus et al.,
2003), only 20% of the CNR glaciers are covered. Hence, the SRTM C-band DEM was used in this
study for glacier surface elevation change. The unfilled finished SRTM C-band DEM is freely available





on *http://earthexplorer.usgs.gov/*. The spatial resolution of SRTM C-band DEM is 1 arc-second (approximately 30 m) and geographic projection is WGS84/EGM96. While the penetration depth of the SRTM C-band radar beam into snow and ice is a critical issue when the SRTM DEM is used for geodetic mass-balance calculations. The elevation difference between the SRTM C-band and X-band DEM can be considered as a first approximation for the penetration depth of the SRTM C-band (Gardelle et al., 2012a; K äb et al., 2012).

### 3.3 TerraSAR-X/TanDEM-X

TerraSAR-X was launched in June 2007, and then its twin satellite, TanDEM-X was launched in June 2010 by the German Aerospace Center (DLR). Fly in close orbit formation, the two satellites acting as a flexible single-pass SAR interferometer (Krieger et al., 2007). Four pairs of X-band bistatic TerraSAR-X/TanDEM-X data points in the experimental Co-registered Single look Slant range Complex (CoSSC) format, acquired in bistatic InSAR stripmap mode, were used in this study (Fig. 1, Tables 1, 2). The frame sizes of these images were approximately $40 \times 60$ km, with resolutions of approximately 2.5 m in both the ground range and azimuth direction. To avoid seasonal variations induced by melting and snow cover, images were chosen mainly from those taken in February or adjacent months. Images were processed separately in interferometric steps and then mosaicked (Werner et al., 2000).

### 3.4 Landsat images

The relationship between glacier mass balance and changes in glacier extent is worth studying. The present glacier outlines were generated from Landsat images. It is best that Landsat images be acquired in the same year as the SRTM and TerraSAR-X/TanDEM-X data. Unfortunately, due to the influence of the Indian monsoon, the CNR was almost permanently covered by snow and cloud, so higher quality images could not be acquired in 2000–2010. The Operational Land Imager (OLI) sensor, on board Landsat-8, provides an excellent new mid-resolution image source for compiling regional-scale glacier inventories and can provide good-quality multispectral images. Acquired from the United States Geological Survey (USGS), the Landsat OLI images are orthorectified with the SRTM, and almost no horizontal shift was observed.

### 4 Methods

### 4.1 Glacier Delineation

Based on scanned and well-georeferenced topographical maps, the outlines of glaciers in the CNR in 1968 were digitized manually. And then the outlines were validated by reference to the original aerial photographs.

Glacier outlines in 2016 were delineated using a ratio threshold method, a division of the visible or near-infrared band and shortwave infrared band of Landsat OLI images (Paul et al., 2009; Racoviteanu et al., 2009). A $3 \times 3$ median filter was applied to eliminate isolated ice patches < 0.01 km$^2$ (Bolch et al., 2010b; Wu et al., 2016). In order to discriminate proglacial lakes, seasonal snow, supraglacial boulders and debris-covered ice, scenes without snow, or cloud-free image scenes acquired at nearly the same time, were used for reference when making manual adjustments. Generated from the SRTM-C DEM automatically, topographical ridgelines (TRLs) were used to divided the final contiguous ice coverage into individual glacier polygons (Guo et al., 2015).

Uncertainty in the glacier outlines arises from positional and processing errors associated with





glacier delineation (Bolch et al., 2010a; Racoviteanu et al., 2008). No distinct horizontal shift was observed in Landsat images and the impact of seasonal snow, cloud and debris cover was eliminated manually (Bolch et al., 2010a; Guo et al., 2015). The best way to assess processing errors is to compare our results with independently digitized glacier outlines from high-resolution air photos (Bolch et al., 2010a; Paul et al., 2003). Compared Landsat-image outlines with real-time kinematic differential GPS (RTK-DGPS) measurements and Google EarthTM images, average offsets of ±10 m and ±30 m were acquired for the delineation of clean and debris-covered ice (Guo et al., 2015), whereas average offsets between topographic-maps outlines and Corona images was ±6.8 m (Wu et al., 2016). Hence, mean relative errors of ±0.8% and ±3.0% were determined for glacier areas in 1968 and 2016, respectively.

## 4.2 Glacier Length

The length of the glacier centreline, a key parameter in glacier inventory, is a most important one for modelling future glacier evolution (Le Bris and Paul, 2013). It has been digitized manually in traditional studies, but this method is inefficient and cannot be replicated. With the rapid development of Geographic Information System (GIS) technology, a few automated techniques have been proposed (Kienholz et al., 2014; Koblet et al., 2010; Le Bris and Paul, 2013; Schiefer et al., 2008). Using a hydrological approach, Schiefer et al. (2008) derived a line representing the maximum flow path water would take over the glacier surface, but these lengths are 10–15% longer than distances measured along actual centrelines. Le Bris and Paul (2013) presented an alternative method, based on a so-called "glacier axis" concept, following a centreline from the highest to the lowest glacier elevation. A limitation of this approach is that the derived line does not necessarily represent the longest glacier centreline or that of the main branch. Kienholz et al. (2014) suggested another method based on a "cost grid—least-cost route approach"; however, it is quite complicated to calculate and some lines have to be adjusted manually.

A new strategy is implemented here (X. Yao et al., 2015) based on a glacier-axis concept derived from glacier morphology that only requires glacier outlines and a digital elevation model (DEM) as input. From GIS modelling techniques, an automatic method is applied to derive the heads, termini and centrelines of glaciers. First, the heads and termini are identified for every glacier. Second, the glacier outline is divided into two curved lines based on its head and terminus. Third, using the method of Euclidean distance, the glacier polygon is divided into two regions. The common boundary of these two regions can be referred to the glacier centreline. This method was applied in the Kangri Karpo Mountains and error estimation was performed by comparing the results with high-resolution aerial imagery at the terminus (Paul et al., 2015). The uncertainties were no more than 6 and 7.5 m in 1968 and 2016, respectively.

## 4.3 Glacier elevation changes

Bistatic interferograms contain both flat earth and topographic phases from which glacier-elevation changes can be derived (Li and Lin, 2017; Li et al., 2018; Neckel et al., 2013; Paul et al., 2015). Two methods can be used. The first, based on differential SAR interferometry (DInSAR), uses orbital information from bistatic SAR images and reference DEMs (here SRTM DEM and TOPO DEM) to simulate the flat earth and topographic phases, and then removes them from the original bistatic interferogram to leave a differential interferogram. The second, common DEM differencing, generates a new DEM from bistatic SAR images, based on InSAR technology, and then performs common DEM differencing with respect to reference DEMs (Neckel et al., 2013). In the DInSAR





method, most parts of the topographic phase have been simulated and removed and the reliability of phase unwrapping increased by the smaller phase gradients (Neckel et al., 2013) so the topographic residual phase can be transformed directly to an elevation change. The DInSAR and common DEM differencing methods were used to detect glacier-elevation changes in the CNR between 1968 and ~2013.

To improve the phase-unwrapping procedure and minimize errors, the unfilled finished SRTM C-band DEM was employed. The use of the DInSAR method to acquire elevation changes from bistatic SAR images can be described by

$$\Delta\varphi_{elevation} \;=\; -\frac{2\pi B_\perp \Delta h}{\lambda R \sin\theta} \;=\; -\left(\frac{2\pi B_\perp \Delta h_{srtm}}{\lambda R \sin\theta} + \frac{2\pi B_\perp \Delta h_{residual}}{\lambda R \sin\theta}\right) \tag{1}$$

where $B_\perp$ is the perpendicular baseline, $\lambda$ is the wavelength of the radar signal, R is geometric distance from the satellite to the scatterer, $\theta$ is the incidence angle, and $\Delta h$ is elevation, which can be split into elevation in SRTM C-band DEM ($\Delta h_{srtm}$) and the elevation changes ($\Delta h_{residual}$) due to glacier thinning or thickening (Kubanek et al., 2015; Li et al., 2018).

It is assumed that no height change occurs in the off-glacier regions. Precise horizontal offset registration between the SRTM C-band DEM and the TerraSAR-X/TanDEM-X acquisitions is mandatory. An initial lookup table was calculated, based on the relationship between the map coordinates of the SRTM C-band DEM segment covering the TerraSAR-X/TanDEM-X master file and the SAR geometry of the respective master file. Due to the side-looking geometry of TerraSAR-X/TanDEM-X, distortion in the foreshortening, layover and shadow regions, can result in some errors. These distortions induce gaps in the lookup table which were filled by linear interpolation. The horizontal offsets between both datasets were calculated by GAMMA's offset_pwrm module for cross-correlation optimization of the simulated SAR images. The horizontal registration and geocoding lookup table was refined with these offsets and used to translate the SRTM C-band DEM from geographic into SAR coordinates. A differential interferogram was then generated from the TerraSAR-X/TanDEM-X interferogram and the simulated phase of the co-registered SRTM C-band DEM. This was filtered by an adaptive filtering approach and the flattened differential interferogram unwrapped with GAMMA's minimum cost flow (MCF) algorithm. The unwrapped differential phase could be transformed to absolute elevation changes from the computed phase-to-height sensitivity and select ground control points (GCPs) of the off-glacier regions of the SRTM C-band DEM. However, the baseline refinement cannot completely eliminate error, so a residual exists in the differential interferogram. This residual can be regarded as a linear trend estimated by a two-dimensional first-order polynomial fit in off-glacier regions. Using polynomial fitting, the residual was removed from maps of absolute differential heights. Finally, the resulting datasets were translated from SAR coordinates into a metric cartographic coordinate system using the refined geocoding lookup table (Paul et al., 2015).

Common DEM differencing with the TOPO DEM and SRTM C-band DEM was employed to acquire the glacier-elevation change from 1968 to 2000 (Liu et al., 2017; Nuth and Kääb, 2011; Pieczonka et al., 2013; Wei et al., 2015). Based on the relationship between elevation difference, slope and aspect, relative horizontal and vertical distortions between the two datasets were corrected statistically (Nuth and Kääb, 2011). At first, a difference map was constructed with the TOPO DEM and SRTM C-band DEM. Before adjustments, histogram statistics for off-glacier regions showed elevation differences concentrated at 6.73 m. Outliers are usually found around data gaps and near DEM edges and can be excluded using 5% and 95% quantile thresholds based on statistical analysis





(Pieczonka et al., 2013). Then, based on the substantial cosinusoidal relationship between standardized
vertical bias and topographical parameters (slope and aspect), the vertical biases and horizontal
displacements could be rectified simultaneously. The biases, caused by different spatial resolutions
between the two datasets, could be refined using the same relationship between elevation differences
and maximum curvatures for both on- and off-glacier regions (Gardelle et al., 2012a). After these
adjustments, the elevation differences in off-glacier regions were concentrated at -0.24 m. It was
concluded that elevation differences in the off-glacier regions had stabilized after these refinements
making the processed DEMs suitable for estimating changes in the glaciers' mass balance.
**4.4  Penetration depth**
When the SRTM DEM is used for geodetic mass-balance calculations, the penetration depth of the
radar signal into snow and ice has to be considered (Berthier et al., 2006; Gardelle et al., 2012a).
Previous studies indicated that the penetration depth affected by the carrier frequency, the density of
snow and ice, and its water content (Berthier et al., 2006; K ää b et al., 2015). Given that the
TerraSAR-X/TanDEM-X were observed mostly in February, when the SRTM was performed, and the
carrier frequencies of the TerraSAR-X/TanDEM-X and the SRTM X-band satellites are almost the
same, it is assumed that no penetration warranting consideration exists between these two datasets (Li
and Lin, 2017). The elevation difference between SRTM C-band and X-band DEMs can be considered
to be the SRTM C-band radar beam penetration into snow and ice (Gardelle et al., 2012a). The
penetration depth in the off-glacier regions was assumed to be zero as the acquisition dates of SRTM
and our TerraSAR-X/TanDEM-X images avoided the main rainy season (Yang et al., 2013), so
elevation differences between SRTM C-band and X-band DEMs could be evaluated with common
DEM differencing. The result showed that the average penetration depth of the SRTM C-band radar is
1.16 m in the CNR. This value is consistent with previous studies finding an average penetration depth
of 1.1 m in Yigong Tsangpo (Zhou et al., 2018).
**4.5  Mass balance and accuracy estimation**
In order to convert glacier-elevation changes to a geodetic mass balance, the glacier area and
ice/firn/snow density must be considered. The geometric union of the 1968 and ~2013 glacier masks
was used to identify area changes (Li et al., 2012; Neckel et al., 2013). An ice/firn/snow density of 850
31  kg m$^{-3}$, with an uncertainty of 60 kg m$^{-3}$, was applied to assess the water equivalent (w.e.) of mass
changes from elevation differences (Huss, 2013; Li et al., 2018; Wei et al., 2015). As the equilibrium
line altitude (ELA) increases gradually from south to north in the CNR (T. Yao et al., 2012), it is
difficult to separate the ablation and accumulation zones so the density value was applied to both. The
final error in geodetic glacier mass balances results from errors in surface elevation measurements
(Gardelle et al., 2013).
Field measurement of off-glacier elevations is the best way to assess the accuracy of the DEMs
employed in this study. While it is difficult to carry out large-scale GPS real-time kinematic (GPS-RTK)
field measurements in off-glacier region, elevations from the ICESat Geoscience Laser Altimeter
System (GLAS) could be employed for a first accuracy assessment. These data are freely available
from the National Snow and Ice Data Center (NSIDC) (release 634; product GLA14). Surface
elevations of the DEMs were extracted at each ICESat footprint location. To ensure the accuracy of
comparison, ICESat points were removed from the analysis if the elevation difference between GLA14
and multi-source DEMs exceeded 100 m in off-glacier region. A mean and standard deviation of 2.14 ±





1.46 m and 1.95 ± 1.76 m were found for the TOPO and SRTM C-band DEMs, respectively. For the
InSAR-derived TerraSAR-X/TanDEM-X DEM, the GCPs that converted the unwrapped interferogram
into absolute heights were selected from the off-glacier regions of the SRTM C-band DEM; the
accuracy of TerraSAR-X/TanDEM-X DEM are similar to those of the SRTM C-band DEM.

In the process of deriving glacier elevation changes, it is assumed that no height change occurred
in the off-glacier regions from 1968 to ~2013. For an error estimate of the derived surface elevation
changes, the residual elevation differences in off-glacier regions needs to be estimated. The mean
elevation differences (MED) between the final difference maps in off-glacier regions ranged from -1.23
to 1.12 m (Table 3). The standard deviation (SD) in off-glacier regions will probably overestimate the
uncertainty of the larger sample because averaging in larger regions reduces the error. The uncertainty
can be estimated by the standard error of the mean (SE) (Berthier et al., 2010):

$$\text{SE} = \text{SD}/\sqrt{N} \qquad (2)$$

where N is the number of the included pixels. To minimize the effect of autocorrelation, a decorrelation
length based on the spatial resolution is recommended. From previous studies, decorrelations of 600 m
and 200 m were employed for different DEMs with the spatial resolution of 30 m and 10 m (Bolch et al.,
2011; Paul et al., 2015). The overall errors of derived surface-elevation changes can then be estimated
using SE and MED in off-glacier regions:

$$\sigma = \sqrt{\text{MED}^2 + \text{SE}^2} \qquad (3)$$

Finally, the overall mass balance errors were determined using the estimated errors of glacier area
and surface elevation change, and the ice density uncertainty of 60 kg m$^{-3}$ (Neckel et al., 2013).

## 5    Results

### 5.1    Area change

There were 715 glaciers with a total area of 1713.42 ± 51.82 km$^2$ in 2016 in the CNR, with a mean
glacier size about 2.40 ± 0.07 km$^2$ (Fig. 2). While large glaciers dominate the area (those >1 km$^2$
occupy 93.5% of the total area) small glaciers dominate the number (those ≤0.5 km$^2$ occupy 69.4% of
the total number) (Fig. 2a). Area distribution by elevation bands is normal. About 82.2% lies in the
4500–5800 m elevation range, 4.9% is below 4200 m, and only 2.4% above 6200 m. The median
elevation is around 5248 m (Fig. 2b). Kyagqen Glacier, the largest glacier (153.07 ± 0.43 km$^2$) has the
lowest tongue at 2882 m a.s.l. The mean glacier surface slope in the CNR is 22.8°, with most in the
12°-32° range, accounting for 93.5% of total area. Glaciers having a SE, S or E aspect account for 85.5%
of their area (Fig. 2c).

The central Nyainqentanglha Range contains almost 100 glaciers with significant debris cover,
about 203.23 km$^2$ or ~11.9% of the whole ice cover (Fig. 3). Among all the debris-covered glaciers,
there were 12 glaciers with areas of debris cover that exceeded 5 km$^2$. Nalong Glacier has the most in
the CNR, 21.88 km$^2$ or ~28.0% of its area. On Kyagqen Glacier, the debris cover only accounts for 3.2%
of its area. Some 89.4% of the debris-covered area is in the 3700-5100 m elevation range, 7.4% is
below 3700 m, and only 3.2% above 5100 m. The lowest elevation of debris cover (2882 m) coincides
with the lowest limit of Kyagqen Glacier. The upper limit of debris cover (5754 m) is on Glacier
5N225E0033, on the north slope of the CNR.

Comparing the total area of all glaciers in 1968 with that in 2016, ice cover in the CNR has
diminished by 824.32 ± 55.65 km$^2$ (32.5% ± 2.2%) or 0.68% ± 0.05% a$^{-1}$ (Table 4). Small glaciers
shrank the most, but large glaciers dominated the absolute area loss (Fig. 2d). Analysis of glacier



hypsography showed that the ice cover below 2800 m, with an area of 0.32 km$^2$, had disappeared
completely, absolute area loss increased gradually with altitude through the 2800–5200 m a.s.l. range,
then decreased gradually from 5200–6200 m a.s.l., remaining almost unchanged above 6200 m. The
average minimum elevation of the glaciers increased by 258 m, while their median elevation rose about
97 m from 5151 to 5248 m. Disintegration of more glaciers compensated for the disappearance of a
few glaciers so the overall number of glaciers increased. Those that had disappeared were small and
situated at relatively low altitudes.

**5.2 Length change**

When comparing the termini of all glaciers in the CNR from 1968–2016 most had retreated. Based on
different glacier size, slope and aspect, 33 glaciers were selected from all the retreating glaciers for
analysis of length changes (Fig. 4, Table 5). These experienced a mean recession of 1432 m (29.8 m
a$^{-1}$), ranging from 217 m to 8826 m. Glacier 5O281B0567, with a mean recession of 4.5 m a$^{-1}$,
experienced the least, its centreline decreasing from 4742 m to 4525 m. Glacier 5O281B0668 at 183.9
15 m a$^{-1}$ experienced the most, its length decreasing from 18706 m to 9879 m. The terminus elevations of
16 these selected glaciers rose an average of 115 m, varying from 29 m (4590 to 4619 m a.s.l.) to 323 m
(3981 to 4304 m a.s.l.).

**5.3 Mass balance**

Significant glacier surface lowering has been observed in the CNR since 1968 with mass losses tending
to increase during the most recent decade. Glaciers, with an area of 2624.30 km$^2$, experienced a mean
thinning of 24.48 $\pm$ 0.30 m (0.54 $\pm$ 0.05 m a$^{-1}$), or a mean mass deficit of 0.46 $\pm$ 0.04 m w.e. a$^{-1}$,
equivalent to an overall mass loss of 64.21 $\pm$0.54 Gt from 1968 to ~2013. The rate of thinning increased
during the investigated periods. Glaciers thinned by 15.69 $\pm$0.28 m, representing a mean mass loss of
0.42 $\pm$0.05 m w.e. a$^{-1}$ from 1968 to 2000. Surface lowering was 9.24 $\pm$0.79 m with a mean mass loss of
0.60 $\pm$0.20 m w.e. a$^{-1}$ from 2000 to ~2013 (Fig. 5 and Table 6).

Heterogeneous mass balances were detected in the CNR from 1968 to ~2013. Glaciers in south
slope, with an area of 1896.36 $\pm$20.39 km$^2$, experienced a mean mass deficit of 0.46 $\pm$0.02 m w.e. a$^{-1}$
from 1968 to ~2013, with means of 0.42 $\pm$0.05 m w.e. a$^{-1}$ and 0.56 $\pm$0.12 m w.e. a$^{-1}$ for 1968–2000 and
2000–~2013, respectively. Losses of 0.48 $\pm$0.08 m w.e. a$^{-1}$ in north slope were larger slightly than those
in south slope from 1968 to ~2013. Glaciers with an area of 727.94 $\pm$ 4.31 km$^2$ in the former
experienced a mean mass loss of 0.40 $\pm$ 0.04 m w.e. a$^{-1}$ from 1968 to 2000, and then increased
significantly to 0.72 $\pm$0.31 m w.e. a$^{-1}$ from 2000 to ~2013.

Changes varied significantly between the different time intervals and individual glaciers, even for
those in the same basin with a similar climate. Star Glacier (5O282A0103) experienced the largest
mass loss from 1968 to ~2013 (0.66 $\pm$0.02 m w.e. a$^{-1}$) with losses much higher in the later period from
2000 to ~2013 (1.15 $\pm$ 0.12 m w.e. a$^{-1}$). Meanwhile Yenong Glacier (5O281B0668), in the same
drainage basin, experienced the smallest losses (0.26 $\pm$0.02 m w.e. a$^{-1}$) from 1968 to ~2013, with even
less from 1968 to 2000 (0.16 $\pm$0.04 m w.e. a$^{-1}$). Accelerating mass losses occurred on most sample
glaciers, except for one glacier in the 5N225E basin and two in 5O281B, where the loss rate slowed.

**6 Discussion**

**6.1 Uncertainty**



Uncertainty in the delineation of glacier outlines comes from both positional and processing
components (Bolch et al., 2010a). In this study, the accuracy of the outlines was assessed by comparing
our results with independently digitized glacier outlines from high-resolution aerial photography, such
as real-time kinematic differential GPS (RTK-DGPS) measurements, Google EarthTM images with a
spatial resolution better than 1 m, and Corona images. An uncertainty model suggested by Pfeffer et al.
(2014) ($e(s) = k \times e \times S^p$ ($k = 3, e = 0.039, p = 0.7$)) was employed to assess our uncertainty estimate.
The results determined a value of 21.47 km$^2$ for glacier delineation uncertainty in the CNR in 2016.
This value is smaller than our estimate of about 51.82 km$^2$. The main reason for this difference is
probably an underestimation by the uncertainty model for the CNR, where more debris-covered ice and
exposed bedrock, surrounded by an ice cover, exist. Thus, our uncertainty estimate for the delineation
of glaciers study should be reliable.
The error in the derived glacier mass balance can result from both systematic and random
components (Li et al., 2018). The latter comes from the precision of TerraSAR-X/TanDEM-X
acquisitions and SRTM DEMs, as well as the total glacier area measured. The systematic component
includes errors in the seasonal effects and penetration depth.
Since geodetic measurements should determine mass balances corresponding to an integer number
of balance years, the seasonal variance of glacier mass balances needs to be considered (Gardelle et al.,
2012b). In the CNR, maritime (temperate) glaciers develop and receive abundant summer monsoon
precipitation. Most accumulation and melting occur simultaneously in the summer.
TerraSAR-X/TanDEM-X and SRTM DEMs are usually acquired in February. To evaluate seasonal
effects, glacier mass budgets determined from TOPO DEMs should be adjusted to the state of glaciers
in February based on mass balance variations. The annual distribution of mass balance is difficult to
establish in the study area because field measurements are lacking. It has been assumed, conservatively,
that precipitation is totally converted into mass accumulation. Based on the Dataset of Daily 0.5 ° × 0.5 °
Grid-based Precipitation in China (V2.0), that for February to April in 1968 was 321 mm, which could
create errors of up to ~0.01 m w.e. a$^{-1}$ for the mass balances of 1968–2000 and 1968–~2013. Compared
to other factors in this study, any errors arising from the seasonal variance of mass balances can be
considered negligible.
Another critical unknown is C-band radar penetration into snow and ice when SRTM C-band
DEMs are used for geodetic mass-balance calculations. Penetration depths of 2.1–4.7 m at 10 GHz
were measured in the Antarctic (Davis and Poznyak, 1993), where the depth decreases as the
temperature and water content of the surface snow increases (Surdyk, 2002). Glaciers in the CNR are
predominantly influenced by the monsoon and have higher snow moisture and temperatures than the
Antarctic (Shi and Liu, 2000). Hence, the assumption is that the penetration of the X-band radar is
small, 1.16 m as estimated by comparing SRTM C- and X-band DEMs in this study. This value is
slightly smaller than that of Gardelle et al. (2013), however, its possible penetration can be considered.
The correction for C-band radar penetration led to average mass changes of +0.03 m w.e. a$^{-1}$ for 1968–
2000 and -0.08 m w.e. a$^{-1}$ for 2000–~2013.

**6.2  Glacier inventory and shrinkage**

The median elevation of a glacier is widely used to estimate the long-term Equilibrium Line
Altitude (ELA) (Braithwaite and Raper, 2010), and is suitable for analysing the governing climatic
conditions (Ke et al., 2016). Heterogeneous median elevations were detected in the CNR, and the
spatial distribution of them reflects their climate dependence. This study area is located north of the





Yarlung Tsangpo River where an important moisture transport path of the Indian monsoon enters the plateau. From the Great Bend of the Yarlung Zangbo, the median elevation increases in SE-NW directions (Fig. 3). On the SE slope of the CNR, the Indian monsoon brings abundant moisture, resulting in a relatively maritime climate and a lower median glacier elevation (below 5000 m). Because the high mountain ranges block water vapour transport to the leeward side, a higher median glacier elevation (above 5300 m) is found on the NW slope. Conversely, the amount of debris cover for the median elevation classes decreases from 25.2% on the SE slope (median elevation <4500 m) to only 2.9% on the NW slope (median elevation >5500 m). The main reasons for this are probably an intensive debris supply from the steep rock walls facing south, the different geology of the SE and NW slopes, and autocorrelation effects between glaciers and the debris cover (Frey et al., 2012; Kääb, 2005; Ke et al., 2016).

Due to the influence of the Indian monsoon, the CNR was almost permanently covered by snow and cloud, so the higher-quality optical-satellite images were rarely available from 2000–2010. To preserve the temporal consistency of the second Chinese glacier inventory (CGI-2, based on Landsat images acquired mainly from 2006–2010), a glacier inventory of the CNR was deliberately omitted. Apart from the CGI-2 there have been some other glacier inventories in the CNR. Using 356 Landsat ETM+ scenes in 226 path-row sets from 1999–2003, Nuimura et al. (2015) compiled the GAMDAM inventory (Glacier Area Mapping for Discharge from the Asian Mountains). There was a larger discrepancy between this and the 2000 Chinese glacier inventory in the western Nyainqentanglha Range, probably because the GAMDAM inventory excluded thin ice on headwalls, the effects of shadow and seasonal snow cover, and tended to include smaller areas than those recommended by the GLIMS guidelines (Arendt et al., 2015; Wu et al., 2016). An improved glacier inventory of the SE Tibetan Plateau (SETPGI) was compiled from Landsat images acquired from 2011–2013, coherence images from ALOS/PALSAR images and the SRTM DEM (Ke et al., 2016). Comparing the SETPGI with our 2016 inventory, a slight discrepancy of 3.7% was found which can be accounted for by a change in glacier area of -0.62% a$^{-1}$ from ~2010 to 2016. Ji et al. (2014, 2015) assessed areal changes for seven glaciers in the CNR (Star, Maguo Lung, Ruoguo, Jiangpu, Nalong, Cape, North Cape and Yangbiegong) from aerial photos and Landsat images acquired between 1968 and 2011. Studies showed the ice cover in the CNR had diminished by about 38.2% (1.23% a$^{-1}$) and 9.8% (0.82% a$^{-1}$) from 1970–1999 and 1999–2011 (Table 7). The glaciers in the study area have shrunk continuously since 1968, although the rate has eased during the most recent decade.

The ice cover in the CNR was reduced by about 0.68% $\pm$ 0.05% a$^{-1}$ between 1968 and 2016. Compared with the recession of mountain glaciers in western China, those in the study area have experienced very strong retreat rates. Except for the Altay (0.75% a$^{-1}$) and Kangri Karpo Mountains (0.71% a$^{-1}$) (Paul et al., 2015; X. Yao et al., 2012), glacier shrinkage in the CNR has been larger than in any other region of western China (Table 7).

**6.3 Changes of glacier elevation and mass balance**

Previous studies agreed that glaciers in the CNR were losing mass, although the results did differ from each other. Based on SRTM and SPOT5 DEMs (24 November 2011), a mean thinning of 0.39 $\pm$ 0.16 m a$^{-1}$ was found by Gardelle et al. (2013), whereas different rates of 1.34 $\pm$ 0.29 m a$^{-1}$, 0.81 $\pm$ 0.32 m a$^{-1}$ and 0.30 $\pm$ 0.13 m a$^{-1}$ from 2003–2009 were recorded by Kääb et al. (2015), Neckel et al. (2014) and Gardner et al. (2013), respectively, using ICESat and SRTM. In this study, SRTM DEM and TerraSAR-X/TanDEM-X acquisitions yielded a mean mass thinning of 0.54 $\pm$ 0.05 m a$^{-1}$ from 2000 to





~2013. Different estimates of SRTM C-band penetration have resulted in thinnings at variance with
those determined by Kääb et al. (2015). An average SRTM C-band penetration of 8–10 m (7–9 m when
based on winter trends assumed to reflect February conditions) was used for the eastern Nyainqentanglha
Mountains in the Kääb et al. (2015) study, much greater than the 1.16 m assumed in this study. Previous
studies suggested an average penetration of 2.4 m in Bhutan and 1.4 m around the Everest (Gardelle et al.,
2013), 2.26 m for clean ice below 6000 m in the Mt. Everest region (Li et al., 2018), 2.5 ±0.5 m for a
wider area including east Nepal and Bhutan (Kääb et al., 2012), 1.67 ± 0.53 m in the western
Nyainqentanglha Mountains (Li and Lin, 2017), and 1.24 m in the Kangri Karpo Mountains (Wu et al.,
2018). Because the CNR lies in the centre of the eastern Himalaya, the western Nyainqentanglha
Mountains and the Kangri Karpo Mountains, the glacier characteristics are similar (Shi and Liu, 2000).
Since penetration depth varies with temperature and water content (Surdyk, 2002), an average
penetration of 1.16 m, in agreement with previous studies, was deemed acceptable and suitable for
estimating glacier elevation changes in the CNR.

Brun et al. (2017) recorded a mean mass deficit of 0.62 ±0.23 m w.e. $a^{-1}$ between 2000 and 2016 in
the Nyainqentanglha Range, based on ASTER optical satellite stereo pairs. Because this result relies
exclusively on satellite optical data, it is not affected by signal penetration. Our determination of a mean
mass loss of 0.60 ±0.20 m w.e. $a^{-1}$ from 2000 to ~2013 in the CNR agrees with that Brun et al. (2017) and
suggests our results are reliable. A larger discrepancy is noted with the Zhou et al. (2018) results in
Yigong Tsangpo from declassified KH-9 images (21 December 1975) and SRTM DEMs; 0.11 ±0.14 m
w.e. $a^{-1}$ vs. our result of 0.42 ±0.05 m w.e. $a^{-1}$ from 1968 to 2000. There are likely two reasons for this
discrepancy: first, a difference in dates of the declassified KH-9 images and topographic maps; second, a
difference in the glacier area measured. The Yigong Tsangpo study measured a glacier area of 1055 $km^2$
while ours was 2622.95 $km^2$. Large differences in acquisition dates and glacier areas may result in
significant disparities in the glacier mass balances determined.

Ice in the comparatively flat lower parts of the larger valley glaciers is much thicker than that in the
steep higher glacier reaches, due to the generalized flow law of ice (assumption of perfect plasticity)
(Cuffey and Paterson, 2010). This suggests that large areas of ice may become subject to melting in the
event of climate warming. Whereas the mass-loss patterns on a debris-covered tongue are complicated,
with supraglacial lakes, ice cliffs and a heterogeneous debris cover (Pellicciotti et al., 2015). Although
melting is considered to be less on glacier parts covered in debris, due to its insulating effect (Benn and
Lehmkuhl, 2000), the surface properties may only have a limited influence on the melt: Thinning was
noticeably greater on the debris-covered ice than the clean ice in the 2800–5700 m a.s.l. range from
1968–~2013 in the CNR (-0.92 ±0.05 m $a^{-1}$ vs. -0.51 ±0.05 m $a^{-1}$) (Fig. 6). Similar results have been
found in the eastern Pamir (Zhang et al., 2016a), the Karakoram (Gardelle et al., 2012b), the western
Himalayas (Berthier et al., 2007; Frey et al., 2012) and the Mt. Everest region (Bolch et al., 2008).

Apart from debris cover, there are other features that may affect surface elevation changes, such as
land- or lake-terminating glaciers, supraglacial lakes, and ice cliffs. Land-terminating glaciers with
heavy debris-covers (5N225E0005, 5N225E0031, Yenong, Xiaqu, Kyagqen, Nalong, Maguolong,
Yangbiegong, Cape and North Cape) experienced a mean thinning of 0.53 ±0.05 m $a^{-1}$ from 1968–~2013,
which was smaller slightly than the regional average (0.54 ± 0.05 m $a^{-1}$). Surface lowering of all
lake-terminating glaciers (5N225E0010, 5N225E0033, Jiongla, Lepu, Daoge, Ruoguo and Star) was
0.62 ± 0.05 m $a^{-1}$, or higher than the regional average. Supraglacial lakes are common on most
debris-covered glaciers but are not expanding as quickly as the proglacial ones (King et al., 2017; Wang
et al., 2013; Ye et al., 2009). There should be a correlation between glacier elevation changes and



supraglacial/proglacial lakes, because the most negative changes of lake-terminating glaciers can be attributed to termini directly affected by the expansion of supraglacial/proglacial lakes (Li et al., 2018; Neckel et al., 2017).

Although ice cliffs account for a small proportion of the total debris-covered area, they can make a disproportionate contribution to total ablation (Benn et al., 2012; Han et al., 2010). On steep slopes, heavy debris slides off leaving very fine debris on the ice cliffs. This reduces the ice albedo so the cliffs absorb more shortwave radiation, which is augmented by longwave radiation from the adjacent warm debris layers (Reid and Brock, 2014). Fig. 7 shows the debris cover on Cape Glacier, its ice cliffs, and supraglacial lake. Compared to glaciers in the Kangri Karpo Mountains, subject to the same climate (Paul et al., 2015), the large debris-covered areas, exposed ice cliffs and supraglacial/proglacial lakes might be one of the reasons for the greater glacier mass loss in the CNR.

### 6.4 Climate change

Based on temperature data from 79 meteorological stations on the Tibetan Plateau (TP), the SE TP was the area with the least warming (Duan et al., 2015). Conversely, the MODIS land surface temperature (MODIS LST) showed that the SE TP experienced the most warming (Yang et al., 2014). The National Centers for Environmental Prediction/National Center for Atmospheric Research (NCEP/NCAR) reanalysis data results indicated a decreasing trend of average annual temperature (You et al., 2010). Similarly with precipitation, a decreasing trend in the SE TP was shown by Global Precipitation Climatology Project (GPCP) data (T. Yao et al., 2012), while a positive trend came from Chinese meteorological station annual precipitation data (Li et al., 2010). Thus, glacier changes in the CNR cannot be related directly to these summaries of climate information.

To analyse the response of glaciers in the CNR to climate change, relevant air temperature and precipitation datasets were taken from the Dataset of Daily $0.5° \times 0.5°$ Grid-based Temperature/ Precipitation in China (V2.0) (Dataset2.0). Dataset2.0 was produced using the thin plate smooth spline method, and a 50 year (1961 to 2010) quality controlled observational daily precipitation data series based on 2472 gauges (http://data.cma.cn/data/cdcindex/cid/00f8a0e6c590ac15.html) for Mainland China. Fig. 8 shows the horizontal distribution of surface temperature and precipitation changes from May to September since 1961. It is clear that increasing surface temperatures and decreasing precipitation have been dominant in the CNR in recent decades. Dataset2.0 shows average precipitation decreasing by more than 40 mm per decade since 1961, resulting in less glacier accumulation. The reduced precipitation on the N slope is smaller than on the S slope, but glaciers on the N slope experienced a more intense mass loss than the S slope. This suggests the influence of precipitation is much less on glacier mass loss in the CNR. The average surface temperature increased by more than 0.2 ℃ per decade in the CNR (with a confidence level <0.05), higher than the rate of global warming (0.12 ℃ per decade, 1951–2012) (IPCC, 2013). The warming rate on the N slope is slightly larger than that on the S slope. Furthermore, a lesser warming rate was present from 1961 to 2000, becoming greater after 2000. The changes of average surface temperature are consistent with the changes of glaciers. The mean mass deficit in the 5O28 drainage basin (on the S slope) was smaller than that in the 5N22 drainage basin (on the N slope) during the investigated periods. Glacier mass loss in the CNR can be attributed to climate warming.

### 7   Conclusion

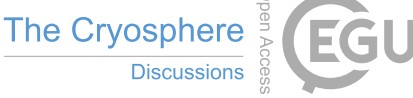



Based on Topographical Maps, Landsat TM/ETM+/OLI images, SRTM and TerraSAR-X/TanDEM-X
acquisitions, the changes of glacier area, length, surface elevation and mass balance in the central
Nyainqentanglha Range during recent decades have been estimated.
Results show that the CNR contained 715 glaciers, with a total area of $1713.42 \pm 51.82$ km$^2$ in
2016. Ice cover has diminished by $0.68\% \pm 0.05\%$ a$^{-1}$ since 1968, but the rate of glacier shrinkage has
lessened during the most recent decade. Compared with the recession of mountain glaciers in western
China, those in the CNR have experienced extremely strong retreat. Overall, the area covered by debris
accounts for 11.9% of the whole ice cover, with the coverage decreasing in a SE-NW directions.
Significant surface lowering of glaciers has been observed since 1968, while mass losses have
tended to increase. Thinning was noticeably greater on the debris-covered ice than the clean ice in the
2800–5700 m a.s.l. altitude range from 1968–~2013. Aside from debris cover, other features affecting
glacier surface elevation changes include land- or lake-terminating glaciers, supraglacial lakes, and ice
cliffs. Based on the Dataset of Daily $0.5\degree \times 0.5\degree$ Grid-based Temperature/Precipitation in China (V2.0),
the glacier mass losses recorded in the CNR can be attributed to climate warming.

Acknowledgements. This work was supported by the fundamental programme of the National Natural Science
Foundation of China (grant no. 41471067), the Ministry of Science and Technology of China (MOST) (grant no.
2013FY111400), the National Natural Science Foundation of China (grant no. 41190084, 41671057, 41671075
and 41701087), the International Partnership Programme of the Chinese Academy of Sciences (grant no.
131C11KYSB20160061) and the grant for talent introduction of Yunnan University. Landsat images are from the
US Geological Survey and NASA. The GAMDAM glacier inventory was provided by A. Sakai. The first and
second glacier inventories were provided by a recent MOST project (2006FY110200). The Dataset of Daily $0.5\degree \times$
$0.5\degree$ Grid-based Temperature/Precipitation in China (V2.0) is from the China Meteorological Data Service Center
(CMDC) in Beijing. All SAR processing was done with GAMMA SAR and interferometric processing software.
We thank DLR for free access to SRTM X-band data and USGS for free access to SRTM C-band and Landsat data.
ASTER GDEM and SRTM are a product of METI and NASA.

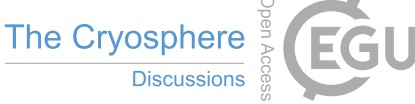

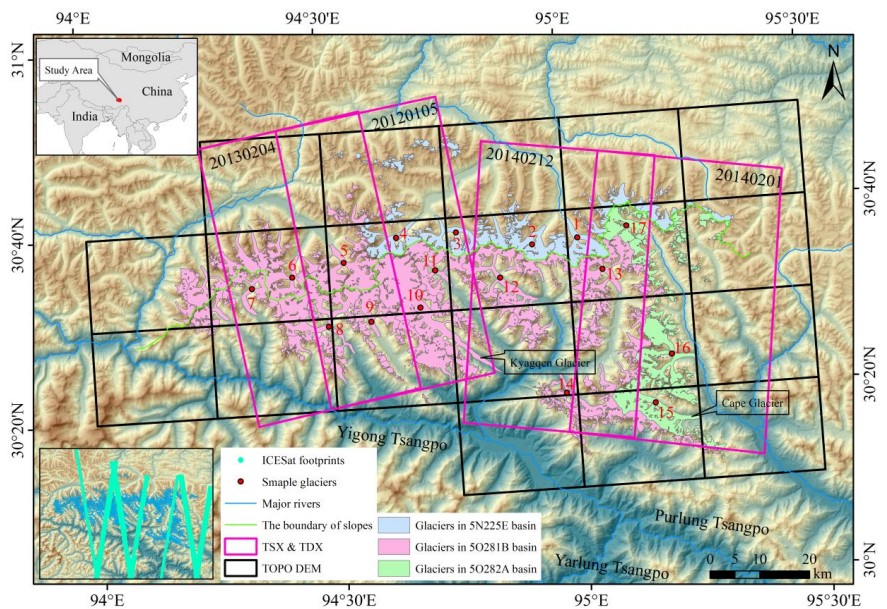

3  Figure 1. Study area and glacier distribution in different drainage basins. TOPO DEMs, TSX/TDX

4  acquisitions and ICESat footprints. Numbers indicate specific sample glaciers chosen for analysis.



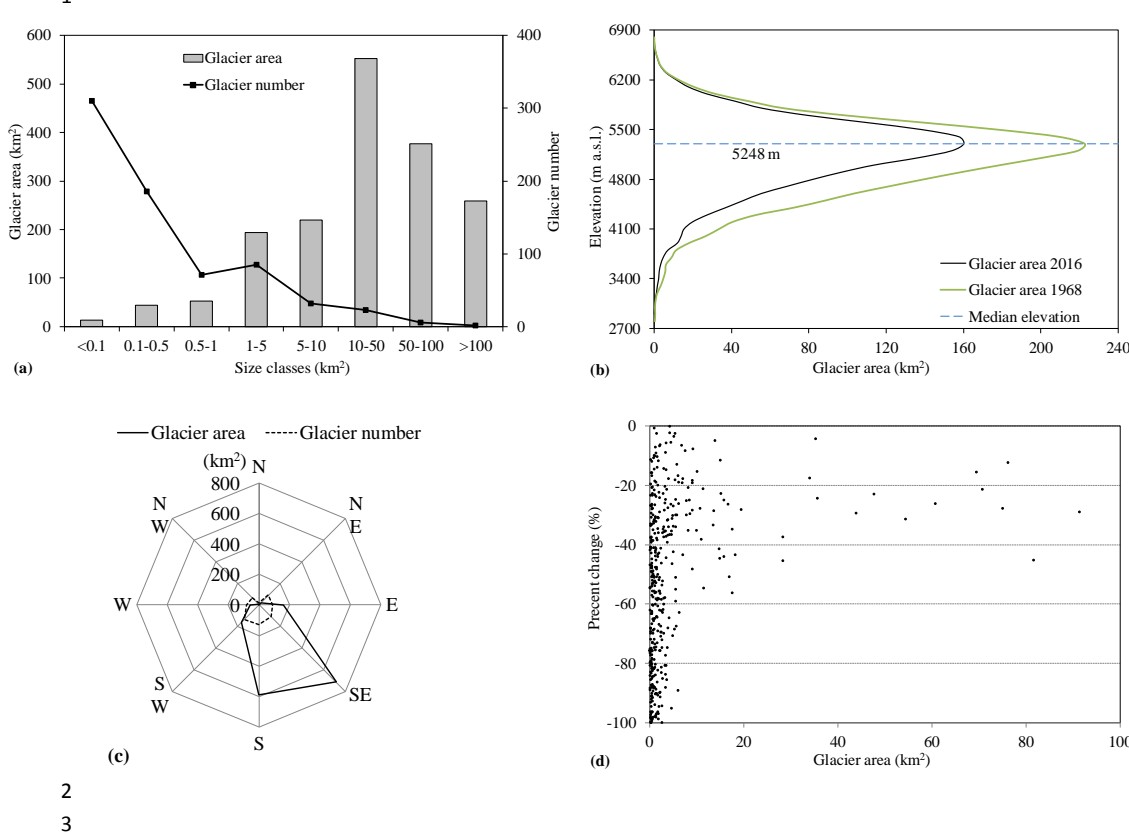

11  Figure 2. Glacier distribution and change in the CNR. (a) Number and area of glaciers in different size

12  categories. (b) Hypsography of glaciers in 1968 and 2016; the dashed line depicts the median elevation

13  value. (c) Number and area of glaciers with different aspects. (d) Percentage change of glacier area

14  from 1968–2016.



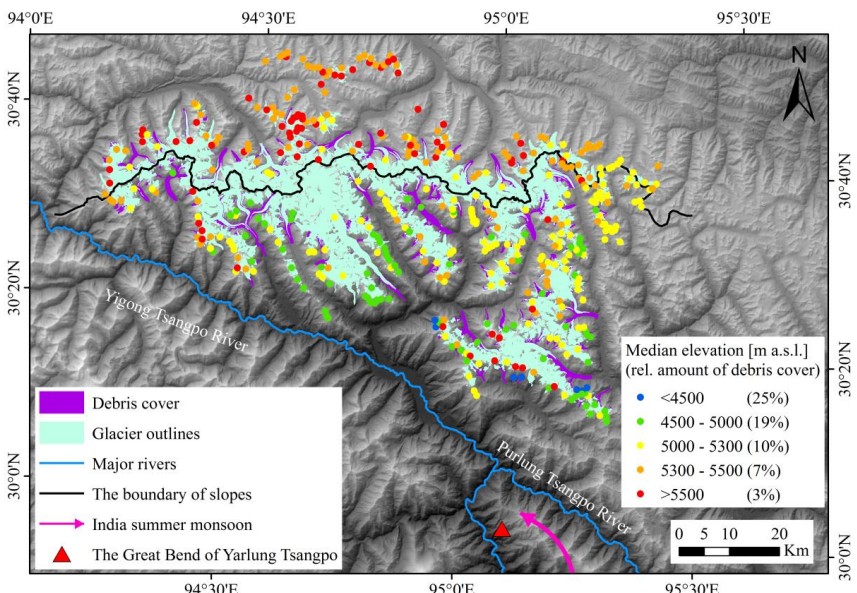

3  Figure 3. Median glacier elevation and relative amount of debris cover is spatially correlated: Median

4  elevation is increasing from southeast to northwest, whereas the debris cover (indicated by the number

5  in brackets in the legend) is decreasing along this gradient.



3    Figure 4. The 33 sample glaciers selected for the generation of centrelines and calculation of length

4    change.





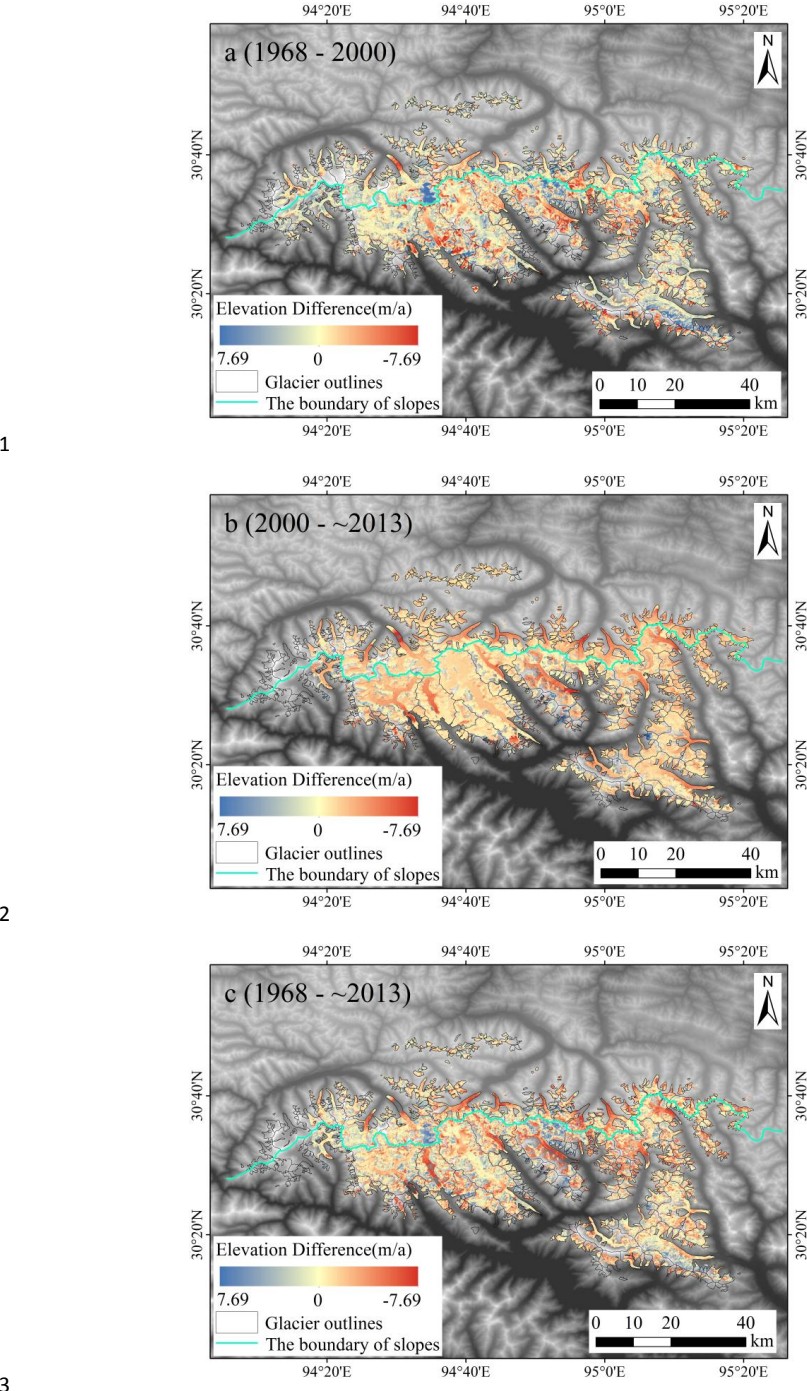

4    Figure 5. Elevation changes in the CNR from 1968 to ~2013. The glacier outlines are based on the

5    geometric union of the 1968 and 2016 glacier extents.



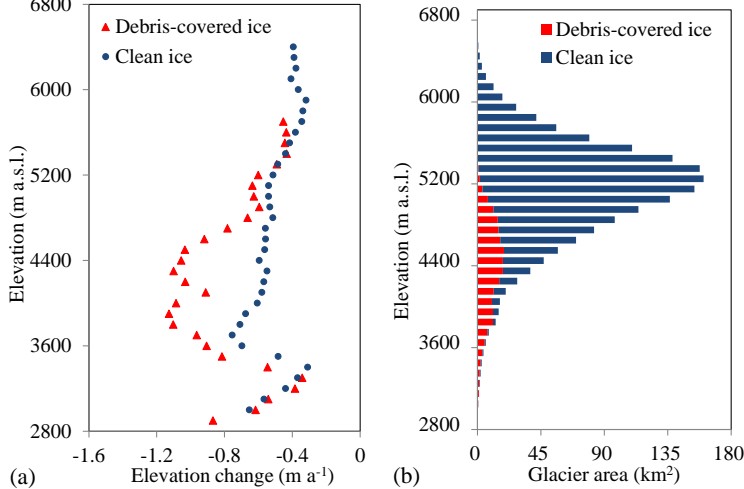

(a)        (b)

4    Figure 6. Glacier elevation changes and distribution of glacier area at each 100 m interval by altitude in

5    the CNR for clean ice and debris-covered ice from 1968 to ~2013.

7



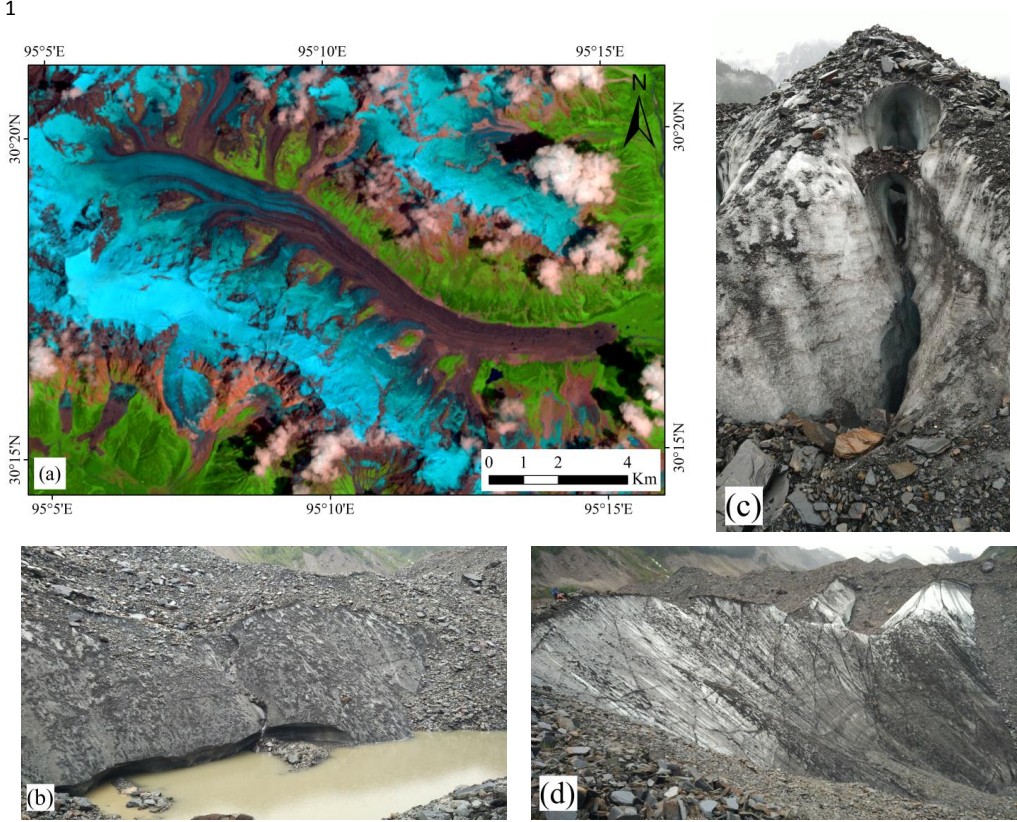

Figure 7. The debris-covered tongue of Cape Glacier with supraglacial lakes and ice cliffs: (a) The

background image is a Landsat OLI image (4 Aug 2013, RGB:743); (b) supraglacial lake; (c)&(d) ice

cliffs (photos taken by K. P. Wu, 12 June 2015).



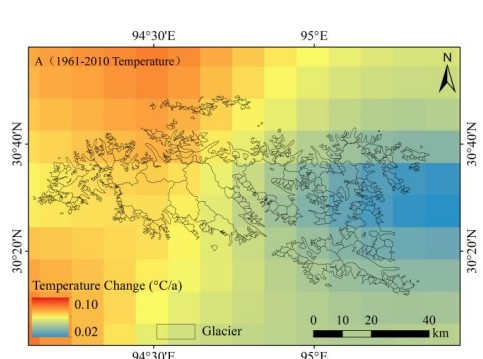 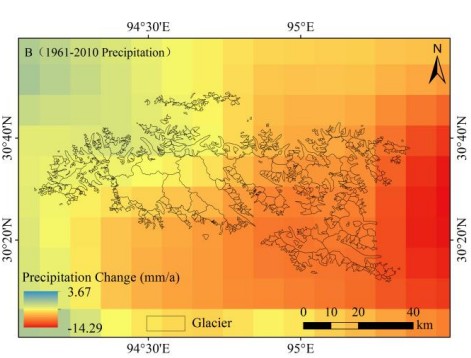

3    Figure 8. The changes of temperature and precipitation (from May to September) in the CNR during

4    1979–2012: (A) Temperature; (B) precipitation.





2    Table 1. Overview of satellite images and data sources.

| Date | Source | ID | Pixel size (m) | Utilisation |
|---|---|---|---|---|
| Apr 1968 | Topographic Maps / TOPO DEM | H46E008018/H46E008019/ H46E008020 H46E008021/H46E008022/ H46E009017 H46E009018/H46E009019/ H46E009020 H46E009021/H46E009022/ H46E010017 H46E010018/H46E010019/ H46E010020 H46E010021/H46E010022 H46E011020 H46E011021/H46E011022 | 12 / 30 | Glacier identification for 1968 and Estimation of glacier elevation change |
| 11–22 Feb 2000 | SRTM C-band | - | 30 | Estimation of glacier elevation change |
| 4 Aug 2013 | Landsat OLI | LC81350392013216LGN00 | 15 | Glacier identification for 2016 |
| 20 Oct 2015 | Landsat OLI | LC81360392015293LGN00 | 15 | |
| 27 Jul 2016 | Landsat OLI | LC81350392016209LGN00 | 15 | |
| 19 Aug 2016 | Landsat OLI | LC81360392016232LGN00 | 15 | |
| 5 Jan 2012 | TSX/TDX | 20120105T114457_20120105T114505 | 12 | Estimation of glacier elevation change |
| 4 Feb 2013 | TSX/TDX | 20130204T114501_20130204T114509 | 12 | |
| 1 Feb 2014 | TSX/TDX | 20140201T233757_20140201T233805 | 12 | |
| 12 Feb 2014 | TSX/TDX | 20140212T233757_20140212T233805 | 12 | |

9    Table 2. Specification of the bistatic TSX/TDX SAR dataset used.

| Date | Relative orbit | Orbital pass | Effective perpendicular baseline (m) | Height of ambiguity (m) | Average incidence angle | Master satellite |
|---|---|---|---|---|---|---|
| 5 Jan 2012 | 67 | Ascending | 94.5 | 88.2 | 47 | TSX |
| 4 Feb 2013 | 67 | Ascending | 126.0 | -64.4 | 45 | TSX |
| 1 Feb 2014 | 59 | Descending | 122.0 | 63.3 | 44 | TSX |
| 12 Feb 2014 | 59 | Descending | 121.2 | 67.1 | 46 | TSX |



3    Table 3. Statistics of vertical errors between the TOPO, SRTM and TSX/TDX.

| Region | Item | MED[1] (m) | STDV[2] (m) | N[3] | SE[4] (m) | $\sigma$[5] (m) |
|---|---|---|---|---|---|---|
| | SRTM - TOPO | -1.22 | 20.40 | 9421 | 0.21 | 1.24 |
| 5N225E Basin | TSX/TDX - SRTM | -1.23 | 6.11 | 84762 | 0.02 | 1.23 |
| | TSX/TDX - TOPO | -0.57 | 26.07 | 84762 | 0.09 | 0.58 |
| | SRTM - TOPO | 0.13 | 27.51 | 14961 | 0.22 | 0.26 |
| 5O281B Basin | TSX/TDX - SRTM | 0.18 | 17.43 | 134579 | 0.05 | 0.19 |
| | TSX/TDX - TOPO | -0.14 | 32.55 | 134579 | 0.09 | 0.17 |
| | SRTM - TOPO | 0.27 | 24.79 | 6591 | 0.31 | 0.41 |
| 5O282A Basin | TSX/TDX - SRTM | 1.12 | 6.81 | 59281 | 0.03 | 1.12 |
| | TSX/TDX - TOPO | 0.26 | 32.51 | 59281 | 0.13 | 0.29 |
| | SRTM - TOPO | -0.24 | 24.76 | 30973 | 0.14 | 0.28 |
| Total | TSX/TDX - SRTM | 0.79 | 12.09 | 278622 | 0.02 | 0.79 |
| | TSX/TDX - TOPO | -0.29 | 30.21 | 278622 | 0.06 | 0.30 |

4    Note: 1 MED - mean elevation difference; 2 STDV - standard deviation; 3 N - the number of considered pixels;

5        4 SE - standard error; 5 $\sigma$ - the overall error of the derived surface elevation change.

9    Table 4. Glacier area changes in the CNR from 1968 to 2016.

| Drainage basin | Glacier area (km²) | | Change of glacier area | |
|---|---|---|---|---|
| | April 1968 | July 2016 | Percent change from 1968-2016 (%) | Average change (% a⁻¹) |
| 5N225E | 316.25±3.63 | 281.45±6.90 | -11.00±2.07 | -0.23±0.05 |
| 5O281B | 1863.99±20.62 | 1167.01±30.71 | -37.39±1.98 | -0.78±0.04 |
| 5O282A | 357.50±3.87 | 264.96±6.32 | -25.89±2.07 | -0.54±0.04 |
| Total | 2537.74±20.28 | 1713.42±51.82 | -32.48±2.19 | -0.68±0.05 |



1 Table 5. Length change of glaciers in the CNR. The uncertainties in glacier length in 1968 and 2016 are

2 6 and 7.5 m. The uncertainties in length change is 0.20 m a$^{-1}$ during 1968 to 2016.

| WGI ID | 1968 | | 2016 | | Length change (m a$^{-1}$) | Rise in terminus elevation (m) |
|---|---|---|---|---|---|---|
| | Length (m) | Terminus elevation (m) | Length (m) | Terminus elevation (m) | | |
| 5N225E0005 | 19465.11 | 4246 | 19002.16 | 4293 | -9.64 | 47 |
| 5N225E0010 | 14915.31 | 4337 | 14010.08 | 4389 | -18.86 | 52 |
| 5N225E0012 | 5895.97 | 4740 | 4226.54 | 4846 | -34.78 | 106 |
| 5N225E0014 | 3354.56 | 5054 | 2791.55 | 5116 | -11.73 | 62 |
| 5N225E0018 | 5819.22 | 4795 | 4587.67 | 4845 | -25.66 | 50 |
| 5N225E0022 | 5201.72 | 4848 | 4164.37 | 4906 | -21.61 | 58 |
| 5N225E0033 | 12624.73 | 4184 | 11642.76 | 4227 | -20.46 | 43 |
| 5N225E0034 | 3074.01 | 5012 | 2457.07 | 5135 | -12.85 | 123 |
| 5N225E0036 | 2907.52 | 4685 | 2600.69 | 4761 | -6.39 | 76 |
| 5O281B0534 | 3355.81 | 4694 | 2240.71 | 4792 | -23.23 | 98 |
| 5O281B0546 | 6083.60 | 4870 | 5309.17 | 4909 | -16.13 | 39 |
| 5O281B0567 | 4742.23 | 4622 | 4525.15 | 4672 | -4.52 | 50 |
| 5O281B0575 | 24703.52 | 3915 | 19657.25 | 4110 | -105.13 | 195 |
| 5O281B0583 | 10774.57 | 4094 | 8506.80 | 4201 | -47.25 | 107 |
| 5O281B0596 | 6781.61 | 4590 | 6501.30 | 4619 | -5.84 | 29 |
| 5O281B0610 | 10804.78 | 4213 | 9773.67 | 4379 | -21.48 | 166 |
| 5O281B0611 | 11068.81 | 4112 | 10521.56 | 4195 | -11.40 | 83 |
| 5O281B0626 | 14162.98 | 4123 | 12339.75 | 4336 | -37.98 | 213 |
| 5O281B0632 | 5734.49 | 4229 | 4483.55 | 4450 | -26.06 | 221 |
| 5O281B0661 | 9636.27 | 3846 | 7707.38 | 4037 | -40.19 | 191 |
| 5O281B0668 | 18706.15 | 3779 | 9879.99 | 3878 | -183.88 | 99 |
| 5O281B0702 | 43458.08 | 3179 | 41656.85 | 3284 | -37.53 | 105 |
| 5O281B0729 | 34651.63 | 2738 | 33993.28 | 2882 | -13.72 | 144 |
| 5O281B0746 | 13687.27 | 3624 | 11756.18 | 3705 | -40.23 | 81 |
| 5O281B0755 | 5487.93 | 3981 | 4332.47 | 4304 | -24.07 | 323 |
| 5O281B0768 | 18371.37 | 3466 | 17789.97 | 3522 | -12.11 | 56 |
| 5O281B0804 | 5312.66 | 3562 | 4250.48 | 3810 | -22.13 | 248 |
| 5O281B0813 | 24408.54 | 3989 | 22164.38 | 4118 | -46.75 | 129 |
| 5O281B0818 | 3323.90 | 4744 | 2461.76 | 4886 | -17.96 | 142 |
| 5O281B0836 | 5143.41 | 4434 | 4331.54 | 4601 | -16.91 | 167 |
| 5O282A0071 | 19662.20 | 3381 | 18471.66 | 3474 | -24.80 | 93 |
| 5O282A0101 | 5840.49 | 4230 | 5277.31 | 4355 | -11.73 | 125 |
| 5O282A0103 | 11395.90 | 4276 | 9871.65 | 4350 | -31.76 | 74 |




Table 6. Mean surface elevation changes and mass balance for single glaciers and different regions in the CNR from 1968 to ~2013. Glacier area is the geometric union of the 1968 and 2016 glacier areas. Mean ΔH is mean surface elevation change and mass balance is the annual mass balance. The glaciers with (*) are lake-terminating glaciers.

| Region | Glacier Code (Glacier Name) | Glacier area (km²) | 1968–2000 | | 2000–~2013 | | 1968–~2013 | |
|---|---|---|---|---|---|---|---|---|
| | | | Mean ΔH (m) | Mass balance (m w.e. a⁻¹) | Mean ΔH (m) | Mass balance (m w.e. a⁻¹) | Mean ΔH (m) | Mass balance (m w.e. a⁻¹) |
| 1 | 5N225E0005 | 44.46 | -22.99 ±1.24 | -0.61 ±0.20 | -17.54 ±1.23 | -1.15 ±0.31 | -40.20 ±0.58 | -0.76 ±0.08 |
| 2 | 5N225E0010* | 39.05 | -29.17 ±1.24 | -0.77 ±0.20 | -18.55 ±1.23 | -1.21 ±0.31 | -43.04 ±0.58 | -0.81 ±0.08 |
| 3 | 5N225E0031 | 46.31 | -12.10 ±1.24 | -0.32 ±0.20 | -11.49 ±1.23 | -0.75 ±0.31 | -23.43 ±0.58 | -0.44 ±0.08 |
| 4 | 5N225E0033* | 38.18 | -29.81 ±1.24 | -0.79 ±0.20 | -10.27 ±1.23 | -0.67 ±0.31 | -39.96 ±0.58 | -0.75 ±0.08 |
| 5N225E basin | | 353.38 | -18.69 ±1.24 | -0.50 ±0.20 | -13.46 ±1.23 | -0.88 ±0.31 | -31.35 ±0.58 | -0.59 ±0.08 |
| 5 | 5O281B0575 (Jiongla)* | 91.53 | -7.22 ±0.26 | -0.19 ±0.04 | -9.30 ±0.19 | -0.61 ±0.05 | -14.23 ±0.17 | -0.27 ±0.02 |
| 6 | 5O281B0583 (Lepu)* | 60.78 | -11.20 ±0.26 | -0.30 ±0.04 | -8.23 ±0.19 | -0.54 ±0.05 | -19.75 ±0.17 | -0.37 ±0.02 |
| 7 | 5O281B0668 (Yenong) | 47.62 | -2.31 ±0.26 | -0.06 ±0.04 | -10.76 ±0.19 | -0.70 ±0.05 | -13.15 ±0.17 | -0.25 ±0.02 |
| 8 | 5O281B0702 (Xiaqu) | 171.17 | -14.43 ±0.26 | -0.38 ±0.04 | -9.57 ±0.19 | -0.63 ±0.05 | -23.37 ±0.17 | -0.44 ±0.02 |
| 9 | 5O281B0714 (Daoge)* | 81.67 | -18.66 ±0.26 | -0.50 ±0.04 | -10.16 ±0.19 | -0.66 ±0.05 | -27.27 ±0.17 | -0.52 ±0.02 |
| 10 | 5O281B0729 (Kyagqen) | 204.67 | -15.78 ±0.26 | -0.42 ±0.04 | -6.04 ±0.19 | -0.40 ±0.05 | -22.01 ±0.17 | -0.42 ±0.02 |
| 11 | 5O281B0746 (Ruoguo)* | 75.06 | -23.82 ±0.26 | -0.63 ±0.04 | -8.13 ±0.19 | -0.53 ±0.05 | -31.31 ±0.17 | -0.59 ±0.02 |
| 12 | 5O281B0768 (Nalong) | 122.72 | -13.06 ±0.26 | -0.35 ±0.04 | -16.07 ±0.19 | -1.05 ±0.05 | -26.37 ±0.17 | -0.50 ±0.02 |
| 13 | 5O281B0813 (Maguolong) | 75.20 | -19.83 ±0.26 | -0.53 ±0.04 | -12.49 ±0.19 | -0.82 ±0.05 | -31.72 ±0.17 | -0.60 ±0.02 |
| 14 | 5O281B0849 (Yangbiegong) | 55.15 | -11.84 ±0.26 | -0.31 ±0.04 | -12.78 ±0.19 | -0.84 ±0.05 | -25.86 ±0.17 | -0.49 ±0.02 |
| 5O281B basin | | 1898.44 | -15.77 ±0.26 | -0.42 ±0.04 | -7.78 ±0.19 | -0.51 ±0.05 | -23.15 ±0.17 | -0.44 ±0.02 |
| 15 | 5O282A0071 (Cape) | 77.07 | -2.55 ±0.41 | -0.07 ±0.07 | -10.41 ±1.12 | -0.68 ±0.29 | -12.53 ±0.29 | -0.24 ±0.04 |
| 16 | 5O282A0083 (North Cape) | 71.78 | -15.66 ±0.41 | -0.42 ±0.07 | -10.80 ±1.12 | -0.71 ±0.29 | -26.53 ±0.29 | -0.50 ±0.04 |
| 17 | 5O282A0103 (Star)* | 47.87 | -17.31 ±0.41 | -0.46 ±0.07 | -16.64 ±1.12 | -1.09 ±0.29 | -32.83 ±0.29 | -0.62 ±0.04 |
| 5O282A basin | | 372.48 | -12.33 ±0.41 | -0.33 ±0.07 | -10.93 ±1.12 | -0.71 ±0.29 | -23.05 ±0.29 | -0.44 ±0.04 |
| Land-terminating glaciers | | 916.13 | -13.92 ±0.28 | -0.37 ±0.05 | -10.59 ±0.79 | -0.69 ±0.20 | -23.95 ±0.30 | -0.45 ±0.04 |
| Lake-terminating glaciers | | 434.14 | -18.28 ±0.28 | -0.49 ±0.05 | -10.97 ±0.79 | -0.72 ±0.20 | -27.71 ±0.30 | -0.52 ±0.04 |
| Debris-covered region | | 203.22 | -17.64 ±0.28 | -0.47 ±0.05 | -25.18 ±0.79 | -1.65 ±0.20 | -41.36 ±0.30 | -0.78 ±0.04 |
| Clean-ice region | | 2421.08 | -15.50 ±0.28 | -0.41 ±0.05 | -7.63 ±0.79 | -0.50 ±0.20 | -22.87 ±0.30 | -0.43 ±0.04 |
| Region of north slope | | 727.94 | -15.01 ±1.24 | -0.40 ±0.20 | -10.95 ±1.23 | -0.72 ±0.31 | -25.17 ±0.58 | -0.48 ±0.08 |
| Region of south slope | | 1896.36 | -15.96 ±0.25 | -0.42 ±0.04 | -8.55 ±0.48 | -0.56 ±0.12 | -24.21 ±0.14 | -0.46 ±0.02 |
| Total | | 2624.30 | -15.69 ±0.28 | -0.42 ±0.05 | -9.24 ±0.79 | -0.60 ±0.20 | -24.48 ±0.30 | -0.46 ±0.04 |



Table 7. A summary of glacier shrinkage in western China. Sample glaciers include Star Glacier, Maguo Lung Glacier, Ruoguo Glacier, Jiangpu Glacier, Nalong Glacier, Cape Glacier, North Cape Glacier and Yangbiegong Glacier.

| Region | | Period | Absolute ice loss (km²) | Area change (%) | Average change (% a⁻¹) | Reference |
|---|---|---|---|---|---|---|
| Altay | | 1960 - 2009 | -104.61 | -36.91 | -0.75 | (X. Yao et al., 2012) |
| Kangri Karpo | | 1980 - 2015 | -679.50 | -24.91 | -0.71 | (Wu et al., 2018) |
| Tian Shan | | 1960 - 2010 | -- | -11.5 | -0.22 | (Wang et al., 2011) |
| eastern Pamir | | 1963 - 2009 | -248.70 | -10.80 | -0.25 | (Zhang et al., 2016) |
| western Kunlun | | 1970 - 2010 | -95.06 | -3.37 | -0.09 | (Bao et al., 2015) |
| Qilian Mountain | | 1956 - 2005 | -417.15 | -20.70 | -0.47 | (Sun et al., 2015) |
| the interior area of the TP | | 1970 - 2009 | -766.65 | -9.54 | -0.26 | (Wei et al., 2014) |
| Everest | | 1976 - 2006 | -501.91 | -15.63 | -0.56 | (Nie et al., 2010) |
| Gangga Mountain | | 1966 - 2009 | -29.20 | -11.33 | -0.28 | (Pan et al., 2012) |
| the CNR | Whole Range | 1968 - 2016 | -824.32 | -32.5 | -0.68 | This study |
| | Sample Glaciers* | 1968 - 1999 | -257.19 | -38.23 | -1.23 | (Ji et al., 2014, 2015) |
| | | 1999 - 2011 | -40.84 | -9.83 | -0.82 | |



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
