# Peer review of "Remote-sensing estimate of glacier mass balance over the central Nyainqentanglha Range during 1968 – ~2013"

_The Cryosphere, 2018_

## Referee Comment (RC1) · Anonymous Referee #1 · 22 Jun 2018

In their TCD manuscript "Remote-sensing estimate of glacier mass balance over the central Nyainqentanglha Range during 1968 – 2013" Wu et al. estimated glacier surface elevation and area changes in the central Nyainqentanglha Range between 1968 and 2013. The authors build their analysis on remote sensing data and older topographic maps. The study has the potential to contribute to our understanding of glacier changes in this area, where glaciological mass balance measurements are lacking. Especially, the long time series to the mid 1960s could be of great value.

However, I have the feeling that many parts of the study are just a methodological replication of an earlier study of the authors (Wu et al., 2018). In its present form the

manuscript does not present any novel concepts, ideas or tools. Having in mind that the methodology is not really new, large parts of the manuscript should be summarized with a reference to the earlier study of the authors. This would certainly increase the readability of the manuscript and the authors could focus on new aspects and interpretation of the data. I further have several aspects concerning the processing and analysis of the data which would certainly involve much interaction of the authors to make the manuscript acceptable for TC.

An idea to increase the novelty of this study would be to process the original aerial imagery (to which the authors state that they have access) with new software solutions to obtain better surface elevations and ortho-images from the mid 1960s (see for example Magnússon et al., 2016).

I will not correct for all mistakes in the English language as there are too many. The manuscript needs to be corrected by a native English speaker before resubmission.

Specific comments:

Abstract: page 1 line 16: please change "those" to glaciers.

Abstract: page 1 line 17-21: this sentence sounds awkward and could be separated into two sentences, one dealing with glacier area changes and one with glacier elevation changes. I further suggest to keep this separation throughout the abstract with a possible synthesis at the end.

Page 2 line 8: typo, hence.

Page 2 line 21-22: please rephrase.

Page 2 line 24-27: what do the authors mean by this sentence? Please rephrase.

Page 2 line 27-31: I think this paragraph is not really necessary in this place.

Page 2 line 29: not only bistatic data is available from the TanDEM-X mission, there are also monostatic acquisitions.

Page 2 line 37-39: please rephrase.

Page 2 line 11-39: I am wondering why the authors did not include the recent and important study of Brun et al., 2017 in their introduction.

Page 2 line 40-Page 3 line 1: this paragraph could start with: In this study, we... and should be better organized. Here the authors should mention the research questions they want to address and the anticipated way to get there.

Page 3 line 3-4: please provide geographic coordinates here as many readers of TC are probably not familiar with Chinese counties.

Page 3 line 6-23: I am wondering if the authors are aware of the study of Loibl et al., 2014 who give a thorough overview of the study area.

Page 3 line 32-34: please rephrase.

Page 4 line 2-4: rephrase.

Page 4 line 4-6: this statement is quite outdated and should be formulated more carefully or even removed here, see for example Berthier et al., 2018 on this issue.

Page 4 line 10-12: language.

Page 4 line 13: what data points? Please rephrase.

Page 4 line 16-18: good point.

Page 4 line 18-19: I think this sentence is more related to data processing. I suggest to keep the data and processing sections clearly separated or to generate one Data & Methods chapter which includes each data source and the associated processing strategy in the same subsection. I leave this to the authors but it should be clearly structured in any way.

Page 4 line 22: yes, but why? This could be moved to the introduction.

Page 4 line 21-30: so in the end only Landsat-8 data is used? Not clear.

Page 4 line 29: almost? Quantify or remove.

Page 4 line 34-35: these lines indicate that the authors have access to the original aerial images. So why not using them directly? Did the authors try to process the data? If this is done successfully it would clearly raise the novelty and impact of the study. For inspiring purposes I suggest to read Magnússon et al., 2016 and Korsgaard et al., 2016.

Page 5 line 5-9: this paragraph is not clear to me. Please rephrase.

Page 5 line 25: is this really a new method implemented in this study? It sounds very similar to the one used in a previous study of the authors (Wu et al., 2018).

Page 5 line 36-Page 7 line 8: I am also wondering in how far this processing scheme differs to the one applied by Wu et al., 2018? I think it would be better to explain the differences to Wu et al, 2018 rather than rewriting everything. This applies not only for this section but for large parts of the manuscript.

Page 7 line 12: true, but the cited studies are rather old and there are more recent studies dealing with this topic, focusing explicitly on TanDEM-X and SRTM data (see for example Vijay, S. et al., 2016 and Neelmeijer, J. et al., 2017).

Page 7 line 14-18: this is probably a reasonable assumption, however surface properties could still have been different in both years.

Page 7 line 23-24: was this mean value used for a correction? Not clear. I strongly suggest to read the above studies (Vijay, S. et al., 2016 and Neelmeijer, J. et al., 2017) as radar penetration is clearly a function of altitude and surface properties. The authors need to account for this.

Page 7 line 32-34: I think this sentence can be removed.

Page 7 line 34-36: so how is it calculated then? Is this sentence related to the following section?

Page 8 line 1-4: I have the feeling that this sentence is rather related to DEM processing than to accuracy assessment. How did the authors find that the accuracy of both DEMs is similar? Not clear.

Page 8 line 10: it might also be interesting to see the sinusoidal relationship between vertical bias and aspect (Page 6 line 36-Page 7 line 3, before and after the correction). Further it would be interesting if any systematic bias is remaining in the off-glacier regions. See for example Neelmeijer, J. et al., 2017 on this issue.

Page 8 line 5-9: true, the elevation difference in off-glacier regions is an important quantity and gives an important insight into the quality of the dataset. I therefore suggest to show the elevation differences in off-glacier regions and do not clip the difference maps (e.g. Fig. 5).

Page 8 line 24-25: is mean glacier size an important quantity (at least the way it is calculated here)? Please explain or remove.

Page 8 line 32: which area? Not clear.

Page 9 line 12-15: language.

Page 9 line 30: not really if error bars are considered.

Page 9 line 31-33: wording.

Page 10 line 29-38: see also my above comments on this issue. There is also an interesting study of Rignot et al., 2001 who discusses differences in penetration depth.

Page 11 line 26-28: and what did they find? Please explain or remove. There is also a recent study of Ji et al., 2018 which might be of interest in this context. I am not sure if Table 7 is really necessary. I have the feeling that it makes things more confusing.

Page 11 line 44: please reconsider the term "mass thinning" and the associated unit.

Page 12 line 10-13: please see my comments above regarding radar penetration depth.

[Figure]

Page 12 line 18-24: a third reason could be a different solution of filling no-data voids in the accumulation areas. While Zhou et al., 2018 use reasonable assumptions to fill no-data gaps, in this study interpolation artifacts might arise from the interpolation of the TOPO DEM.

Figure 3: I assume that the colored dots represent the median elevation of each glacier, but how did the authors calculate the relative amount of debris cover? E.g. has every glacier with a yellow dot a relative debris cover of 10%? Please explain.

Page 12 line 44-Page 13 line 3: did the authors find such a correlation?

Page 13 line 13-41: I am not a climate modeler, however when looking at Figure 8 I am wondering how well this dataset is resolved in mountainous areas such as the study area and how well this dataset fits to other estimates in the region. I think such points need to be discussed in more detail. Are there any other glacier related studies relying on this dataset?

Page 14 line 5-6: I am wondering on which data basis the authors draw this conclusion, having only two time steps of glacier area estimates (1968 and 2016).

Figure 5: I think that at this scale the reader is not able to detect any details in the elevation change maps. I therefore suggest to additionally zoom in into several sub-sections which could be shown in a supplement. As mentioned before it would be of great value to also see the elevation changes in the off-glacier regions. I also found the limits of the color bar rather strange. Is 7.69 m really the maximum value? Furthermore I have the feeling that the TOPO DEM has several regions with unreliable interpolation artifacts. As mentioned above, generating DEMs from the original aerial images could possibly improve the results.

Table 5 and 6: here the reader is not able to locate the glaciers in the Figures. There are numbers in Figure 1 which also could be used here to link the tables with the figures. These numbers could further be shown in Figure 5.
Additional References:

Berthier, E., Larsen, C., Durkin, W. J., Willis, M. J., and Pritchard, M. E. (2018): Brief communication: Unabated wastage of the Juneau and Stikine ice-fields (southeast Alaska) in the early 21st century, The Cryosphere, 12, 1523-1530, https://doi.org/10.5194/tc-12-1523-2018.

Korsgaard, NJ and 6 others (2016): Digital elevation model and orthophotographs of Greenland based on aerial photographs from 1978–1987. Sci. Data, 3, 160032 (doi: 10.1038/sdata.2016.32)

Ji, Q., Yang, T., Dong, J. and He, Y. (2018): Glacier variations in response to climate change in the eastern Nyainqêntanglha Range, Tibetan Plateau from 1999 to 2015, Arctic, Antarctic, and Alpine Research, 50:1, DOI: 10.1080/15230430.2018.1435844.

Loibl, D., Lehmkuhl, F., Grießinger, J., (2014): Reconstructing glacier retreat since the Little Ice Age in SE Tibet by glacier mapping and equilibrium line altitude calculation. Geomorphology 214, 22–39.

Magnússon, E., Muñoz-Cobo Belart, J., Pálsson, F., Ágústsson, H., and Crochet, P. (2016): Geodetic mass balance record with rigorous uncertainty estimates deduced from aerial photographs and lidar data – Case study from Drangajökull ice cap, NW Iceland, The Cryosphere, 10, 159-177, https://doi.org/10.5194/tc-10-159-2016.

Neelmeijer, J., Motagh, M., Bookhagen, B. (2017): High-resolution digital elevation models from single-pass TanDEM-X interferometry over mountainuous region: a case study of Inylchek Glacier, Central Asia. ISPRS J. Photogramm. Remote Sens. 130, 108–121. http://dx.doi.org/10.1016/j.isprsjprs.2017.05.011.

Rignot E, Echelmeyer K and Krabill W (2001): Penetration depth of interferometric synthetic-aperture radar signals in snow and ice. Geophys. Res. Lett., 28(18), 3501–3504. doi: 10.1029/2000GL012484.

Vijay, S.; Braun, M. Elevation Change Rates of Glaciers in the Lahaul-Spiti (Western

[Figure]

Himalaya, India) during 2000–2012 and 2012–2013 (2016): Remote Sens. 2016, 8, 1038.

---

## Author Comment (AC2) · 2 Jul 2018

Dear Referee #1,

Thank you for your valuable suggestions and I have already revised the article according to your suggestions. The following are a few answers to some questions.

General comments: An idea to increase the novelty of this study would be to process the original aerial imagery (to which the authors state that they have access) with new software solutions to obtain better surface elevations and ortho-images from the mid-1960s.

[Figure]

Answer: I am sorry that I did not make it clear due to the mistake in the English language. I didn't have access to process the original aerial imagery but the topographic maps. Topographic maps were compiled by the Chinese Military Geodetic Service from aerial images acquired in April 1968. They were not acquired with new software solution, but the accuracy of topographic maps is very high. According to the photogrammetric Chinese National Standard (2008) issued by the Standardization Administration of the People's Republic of China, the nominal vertical accuracy of these topographic maps is within 3-5 m for flat and hilly areas (with slopes of $< 2°$ and $2-6°$, respectively) and within 8-14 m for the mountainsides and high mountain areas (with slope of $6-25°$ and $>25°$, respectively). The horizontal accuracy of the topographic maps is within 0.5 mm for flat and hilly areas and with 0.75 mm for the mountainsides and high mountain areas. Hence, glacier area and mass balance result from topographic maps should be reliable.

Specific comments:

(1) Page 3 line 6-23: I am wondering if the authors are aware of the study of Loibl et al., 2014 who give a thorough overview of the study area.

Answer: I have already revised the section of study area. "The CNR (30°9′∼ 30°53′N, 94°0′∼ 95°30′E) lies in south-eastern Tibet, north of Linzhi County, east of Jiali County and west of Bomi County, extending about 130 km from west to east. South of this region is the Yigong Tsangpo River, a tributary of the Purlung Tsangpo River and a secondary tributary of the Yarlung Tsangpo River (Fig. 1). The altitudinal differences between mountain peaks and valley bottoms often reach 3000-3500 m. The rugged topography with steep valleys and slopes results from the interplay of a still ongoing tectonic uplift and erosion (Li et al., 1986). High precipitation amounts during the summer monsoon season (May to September) are the main reason for the intense erosion. The CNR is characterized by a strong climatic influence of the Indian summer monsoon entering through the Yarlung Tsangpo valley (Loibl et al., 2014). More than 80% of annual precipitation falls from June to September,

while winter months are characterized by cold and dry conditions (Molnar et al., 2010). According to the climatic classification of local meteorological station data, the CNR marks a transition zone between warm-wet subtropical and cold-dry plateau conditions (Leber et al., 1995). Previous studies showed that average annual precipitation in most high-elevation areas in the CNR exceeds 2000 mm. Farther north, the east-west striking ranges act as barriers forcing heavy orographic rainfalls (Böhner, 2006; Maussion et al., 2014). This results in a distinct precipitation gradient from south toward north slope of the CNR (Shi et al., 1988). Due to the high elevation, the monsoonal summer precipitation accumulates as snow in the upper reaches of the mountain range, a large number of maritime (temperate) glaciers developed (Shi et al., 2008; Shi and Liu, 2000). The first Chinese Glacier Inventory (CGI) determined that glaciers covered 2537.7 km2 of our study region, with a total volume of 454.2 km3 in 1968 (Mi et al., 2002; Pu, 2001); about 8% of the area was covered by debris. Three glaciers in the CNR are larger than 100 km2, the Xiaqu (CGI code: 5O281B0702), Kyagqen (CGI code: 5O281B0729) and Nalong (CGI code: 5O281B0768). The Kyagqen, on the south slope of the CNR, 35.3 km long and 206.7 km2, with a terminus at 2900 m a.s.l., is the largest of these (Li et al., 1986). Above 4000 m a.s.l. it has a broad basin in which several ice streams converge to form a large accumulation zone (165 km2) that accounts for over 80% of the glacier's total area. Below this, the glacier enters a narrow ice-filled valley where its velocity increases; the resultant great driving force pushing the glacier terminus to a subtropical elevation at 2900 m a.s.l. The narrow glacier tongue, 1000 m wide and 17 km long, passes through the subalpine shrub-meadow zone, the mountain dark coniferous forest zone and the mixed broadleaf-conifer forest zone (Li et al., 1986)."

(2) Page 5 line 36-Page 7 line 8: I am also wondering in how far this processing scheme differs to the one applied by Wu et al., 2018? I think it would be better to explain the differences to Wu et al, 2018 rather than rewriting everything. This applies not only for this section but for large parts of the manuscript.

Answer: Actually the data processing in this study was almost simultaneous with that one in Wu et al., 2018. The only difference is that the two-dimensional first-order polynomial fitting in off-glacier regions removes the residual in the differential interferogram. This study area is closed to the study area in Wu et al., 2018, and this processing scheme is similar with previous study, but the Result and Discussion have big differences, such as the distribution of debris covered region, the effect of debris-cover to mass balance, and the difference of land-and lake-terminating glacier mass balance. It is indicated that this study has great scientific value and it is definitely worth to study.

(3) Page 7 line 14-18: this is probably a reasonable assumption, however surface properties could still have been different in both years.

Answer: Surface properties have been different in both years, it is reflected in precipitation in acquired dates of SRTM and TerraSAR-X/TanDEM-X. Given that the SRTM and TerraSAR-X/TanDEM-X were observed mostly in February, winter months are characterized by cold and dry conditions in study area, and the carrier frequencies of the TerraSAR-X/TanDEM-X and the SRTM X-band satellites are almost the same, it is assumed that no penetration warranting consideration exists between these two datasets.

(4) Page 7 line 23-24: was this mean value used for a correction? Not clear. I strongly suggest to read the above studies (Vijay, S. et al., 2016 and Neelmeijer, J. et al., 2017) as radar penetration is clearly a function of altitude and surface properties. The authors need to account for this.

Answer: I have already revised the section of "4.4 Penetration depth". "The penetration depth differences were analysed and corrected in each 100 m elevation bin. Because the penetration difference should not exceed 10 m (Gardelle et al., 2012a), all of the difference values greater than ±10 m were defined as outliers and did not consider for the penetration estimation. The median values of each elevation bin were used to correct the SRTM C-band DEM but only for areas with elevations below 6200 m a.s.l. In the CNR, the penetration depth difference for clean ice/firn/snow was about 0.88 m below

5200 m, and 1.28 m between 5200 and 6200 m. There have no enough pixels per elevation bin were available to generate reliable results at higher elevations. Glacier area above 6200 m occupies about 0.78% of the total glacier area in this study. According to the linear trend calculated for the several highest bins in Fig. 3, a value of 2.26 was assumed for the area above 6200 m. Interestingly, some of the lower elevation bins in debris-covered area show negative correction values. Because the stable areas were also partly covered with snow, the radar penetration difference due to this snow cover was removed during the vertical matching of SRTM C-band and X-band DEMs., Low-elevation bins in debris-covered area have less snow cover than the stable areas, and this will result in negative radar penetration differences. Hence, penetration corrections were not applied for the debris-covered regions. In total, the average penetration depth of the SRTM C-band radar is 1.16 m for clean ice/firn/snow in the CNR. This value is consistent with previous studies finding an average penetration depth of 1.1 m in Yigong Tsangpo (Zhou et al., 2018)."

(5) Page 8 line 1-4: I have the feeling that this sentence is rather related to DEM processing than to accuracy assessment. How did the authors find that the accuracy of both DEMs is similar? Not clear.

Answer: Page 8 line 1-4: "For the InSAR-derived TerraSAR-X/TanDEM-X DEM, the GCPs that converted the unwrapped interferogram into absolute heights were selected from the off-glacier regions of the SRTM C-band DEM; the accuracy of TerraSAR-X/TanDEM-X DEM are similar to those of the SRTM C-band DEM." In section of "4.3 Glacier elevation changes", I have already introduced that "The unwrapped differential phase could be transformed to absolute elevation changes from the computed phase-to-height sensitivity and select ground control points (GCPs) of the off-glacier regions of the SRTM C-band DEM" . It can assume that the off-glacier regions of TerraSAR-X/TanDEM-X DEM are similar to those of SRTM C-band DEM. The accuracy of DEMs were assessed by off-glacier elevations, so the accuracy of TerraSAR-X/TanDEM-X DEM are similar to those of the SRTM C-band DEM.

(6) Page 8 line 10: it might also be interesting to see the sinusoidal relationship between vertical bias and aspect (Page 6 line 36-Page 7 line 3, before and after the correction). Further it would be interesting if any systematic bias is remaining in the off-glacier regions. See for example Neelmeijer, J. et al., 2017 on this issue.

Answer: I have already added the sinusoidal relationship between vertical bias and aspect. "Based on the relationship between elevation difference, slope and aspect, relative horizontal and vertical distortions between the two datasets were corrected statistically (Nuth and Kääb, 2011). At first, a difference map was constructed with the TOPO DEM and SRTM C-band DEM. Before adjustments, histogram statistics for off-glacier regions showed elevation differences concentrated at 6.73 m. Outliers are usually found around data gaps and near DEM edges and can be excluded using 5% and 95% quantile thresholds based on statistical analysis (Pieczonka et al., 2013). Then, based on the substantial cosinusoidal relationship between standardized vertical bias and topographical parameters (slope and aspect), the vertical biases and horizontal displacements could be rectified simultaneously (Fig. 2). The biases, caused by different spatial resolutions between the two datasets, could be refined using the same relationship between elevation differences and maximum curvatures for both on- and off-glacier regions (Gardelle et al., 2012a)." For the systematic bias in the off-glacier regions, the iterations continued until the change of the magnitude of the shift vector was less than 0.3 m.

(7) Page 11 line 26-28: and what did they find? Please explain or remove. There is also a recent study of Ji et al., 2018 which might be of interest in this context. I am not sure if Table 7 is really necessary. I have the feeling that it makes things more confusing.

Answer: In this section, we concluded that the glaciers in the study area have shrunk continuously since 1968 (1.23% a-1 from 1970–1999, 0.82% a-1 from 1999–2011 and 0.62% a-1 from ~2010 to 2016), although the rate has eased during the most recent decade. The study of Ji et al., 2018 present the change of glaciers in whole eastern

Nyainqentanglha Range, the mean glacier size in eastern Nyainqentanglha Range is smaller than that in central Nyainqentanglha Range. In this and previous studies a greater relative loss has been measured for the smaller glaciers. In order to analysis the tendency of glacier area change in CNR, the study of Ji et al., 2018 cannot be discussed in this study. I have already deleted the Table 7.

(8) Figure 3: I assume that the colored dots represent the median elevation of each glacier, but how did the authors calculate the relative amount of debris cover? E.g. has every glacier with a yellow dot a relative debris cover of 10%? Please explain.

Answer: The relative amount of debris cover is the ratio of the area of debris cover to the total glacier area per elevation class.

(9) Page 13 line 13-41: I am not a climate modeler, however when looking at Figure 8 I am wondering how well this dataset is resolved in mountainous areas such as the study area and how well this dataset fits to other estimates in the region. I think such points need to be discussed in more detail. Are there any other glacier related studies relying on this dataset?

Answer: I have already revised this section. "To analyze the response of glaciers in the CNR to climate change, relevant air temperature and precipitation datasets were taken from the Dataset of Daily $0.5° \times 0.5°$ Grid-based Temperature/ Precipitation in China (V2.0) (Dataset2.0). Dataset2.0 was produced using the thin plate smooth spline method, and a 50 year (1961 to 2010) quality controlled observational daily climate data series based on 2472 gauges (http://data.cma.cn/data/cdcindex/cid/00f8a0e6c590ac15.html) for Mainland China. Dataset2.0 is exact describing the climate characteristic of the Tibetan Plateau, the Tienshan Mountains and Tarim Basin (Zhao and Zhu, 2015). Fig. 8 shows the horizontal distribution of surface temperature and precipitation changes from May to September since 1961. It is clear that increasing surface temperatures and decreasing precipitation have been dominant in the CNR in recent decades. The changes in surface temperature and precipitation were confirmed with data from the three nearest meteorological stations, Jiali (30°40′N, 93°17′E, 4488 m a.s.l.), Linzhi (29°40′N, 94°20′E, 2992 m a.s.l.) and Bomi (29°52′N, 95°46′E, 2736 m a.s.l.). Surface temperature at these stations increased slightly from 1961 to 2015, while the trend of precipitation is not evident at the three stations and present large interannual precipitation fluctuations. Dataset2.0 shows average precipitation decreasing by more than 40 mm per decade since 1961, resulting in less glacier accumulation. The reduced precipitation on the N slope is smaller than on the S slope, but glaciers on the N slope experienced a more intense mass loss than the S slope. This suggests the influence of precipitation is much less on glacier mass loss in the CNR. The average surface temperature increased by more than 0.2°C per decade in the CNR (with a confidence level <0.05), higher than the rate of global warming (0.12°C per decade, 1951–2012) (IPCC, 2013). The warming rate on the N slope is slightly larger than that on the S slope. Furthermore, a lesser warming rate was present from 1961 to 2000, becoming greater after 2000. The changes of average surface temperature are consistent with the changes of glaciers. The mean mass deficit in the 5O28 drainage basin (on the S slope) was smaller than that in the 5N22 drainage basin (on the N slope) during the investigated periods. Glacier mass loss in the CNR can be attributed to climate warming."

There have no other glacier related studies relying on this dataset, but this dataset was produced by the Climate Data Center, National Meteorological Information Center, China meteorological Administration and has excellent reliability.

(10) Figure 5: I think that at this scale the reader is not able to detect any details in the elevation change maps. I therefore suggest to additionally zoom in into several subsections which could be shown in a supplement. As mentioned before it would be of great value to also see the elevation changes in the off-glacier regions. I also found the limits of the color bar rather strange. Is 7.69 m really the maximum value? Furthermore I have the feeling that the TOPO DEM has several regions with unreliable

interpolation artifacts. As mentioned above, generating DEMs from the original aerial images could possibly improve the results.

Answer: Thank you for your valuable suggestions and I have already added more details in a supplement, including several subsections and the elevation changes in the off-glacier regions. For the maximum value, elevation differences with values exceeding $\pm100$ m were defined as outliers and omitted, so the 7.69 m a-1 is the maximum elevation difference. As mentioned above, I didn't have access to process the original aerial imagery but the topographic maps. Topographic maps were compiled by the Chinese Military Geodetic Service from aerial images acquired in April 1968. They were not acquired with new software solution, but the accuracy of topographic maps is very high.

Best Regards, Wu Kunpeng and other authors

Please also note the supplement to this comment:
https://www.the-cryosphere-discuss.net/tc-2018-90/tc-2018-90-AC2-supplement.zip

---

## Referee Comment (RC2) · Anonymous Referee #2 · 20 Jul 2018

General comments

This work entitled "Remote-sensing estimate of glacier mass balance over the central Nyainqentanglha Range during 1968 – ∼2013" by Wu et al. estimates glacier mass balance over the central Nyainqentanglaha Range during 1968-2013, using elevation data obtained from topographic maps (1968), SRTM (2000), and TerraSAR-X/TanDEM-X (2013). By measuring two separate intervals, the authors quantify an acceleration of ice loss rates in the region. They also analyze debris-covered glacier thinning relative to clean ice glaciers, and highlight temperature and precipitation trends (at a 0.5 x 0.5 degree resolution) from the Grid-Based temperature precipitation in China V 2.0

Dataset. Based on these trends, the authors suggest that overall temperature changes are consistent with the changing glaciers.

The geodetic mass balance portion of the study is done reasonably well, although some issues should be addressed, particularly regarding how well the given uncertainty ranges capture the true uncertainty involved with using the historical topographic maps.

The climate analysis provided at the end of the paper is very brief, based on a single climate dataset of course spatial resolution, and thus its robustness is somewhat questionable (see specific comments below).

By quantifying the rates of ice loss in the region over several decades, this study has potential to improve our understanding of multi-decade glacier changes and water resources in an important region. However, a primary concern with this paper is the large degree of overlap with a previously published work covering the same topic in a nearby area (https://doi.org/10.5194/tc-12-103-2018). Many aspects of the methodology are nearly identical, except that a 1968 topographic map is used instead of a 1980 topographic map. I recommend the authors summarize all similar aspects, then simply cite their previous work. This will allow the new manuscript to focus on the unique portions, such as the different time intervals and the quantification of the acceleration of ice loss. This will require a significant revision by the authors. However, in its current form I feel the manuscript is too similar to the previous work for publication in TC.

Specific comments:

Page 1 line 26: The uncertainties for the 1968-2000 interval ($\pm$ 0.05) seem rather small, especially when viewing the vertical error statistics in Table 3, where the standard deviation of vertical error between TOPO and SRTM ranges from 20 to 27 meters. Figure 5a (elevation difference between 1968 and 2000) also shows large areas with significant vertical error over both ice and ice-free terrain, which may be due to interpolation procedures used when the topographic map was originally created. The limits

of the color scaling (-7.69 to 7.69 meters) also seem too narrow - a wider elevation range should be used in this figure so that larger elevation changes are not saturated at the endpoints of the color bar. Based on these, I would expect uncertainties larger than $\pm$ 0.05. I recommend careful revisit of the uncertainty estimation procedure, to ensure that the results are representative of the vertical error associated with using the historical topographic maps.

Page 2 line 1: The latest studies by Brun et al. (https://doi.org/10.1038/ngeo2999) and Zhou et al. (https://doi.org/10.1016/j.rse.2018.03.020) are cited later in the paper, but should probably be included here as well.

Page 2 line 40: I am wondering if the authors drew the topographic maps themselves? The wording is unclear here.

Page 3 line 3: Most readers will not be as familiar with the region, thus using county names may not be the best way to describe the location.

Page 4 line 25: Not sure what is meant here by "higher quality images could not be acquired in 2000-2010". There are several Landsat 7 scenes obtained in the early 2000's which are cloud-free and available over the region of study (Landsat scene LE71350392001335BJC00 acquired on Dec 01 2001 for example). What is this referring to?

Page 4 line 29: "No horizontal shift was observed". What is the horizontal shift being measured relative to?

Page 4 line 33: Do the authors have access to the original aerial photographs? If so, it would be useful and interesting to create a figure showing a sample aerial image for some of the glaciers. Also, later in in the paper, it states that Corona satellite images were used to estimate the uncertainty of glacier outlines (Page 10 line 5). Were the Corona images used in the creation of glacier outlines as well? Please clarify exactly how the aerial photographs, topographic maps, and Corona images were used

to derive the glacier outlines.

Page 5 lines 5-9: The logic in this statement is unclear to me. How are the ±0.8% and ±3.0% values derived?

Page 5 lines 25-34: A more detailed and clear description is required to understand how the glacier centerlines were derived. In what way are the glaciers divided into two polygons? Also I do not understand how and for what purpose the derived centerlines are compared with high-resolution aerial imagery. How are the uncertainties of 6 and 7.5 meters obtained? Perhaps a figure helping to illustrate this process would be helpful.

Page 8 line 9: Regarding the statement "probably overestimate the uncertainty of the larger sample". What "larger sample" is being referred to here?

Page 8 line 13: How do the decorrelation lengths factor into the uncertainty estimates? An additional equation showing exactly how they are used would be helpful.

Table 3: Using MED as an abbreviation for mean may be confusing, as MED is commonly used to abbreviate median.

Page 8 Line 33: How were debris-covered portions of the glaciers delineated?

Page 9 line 14: The magnitude of this length change seems extremely large. Is this a lake-terminating glacier? Can the authors show images of the glacier in 1968 and 2016 to confirm this? On another note, it is difficult in this paper to determine which glacier ID corresponds to which glacier in the figures. For example, the numbers in Figure 1 should correspond to the numbers in Table 6, as currently this does not seem to be the case.

Page 13 line 13: It is interesting to see the 0.5 x 0.5 degree temperature and precipitation trends derived from the Grid-based China v 2.0 dataset. However, it seems rather tenuous to base conclusions on a single climate dataset at very course spatial resolution over a region of extreme mountain topography. As the authors summarize in the

paragraph starting on page 13 line 14, different climate datasets give widely varying results. The authors have chosen this particular climate dataset to base their conclusions on - is there sufficient reasoning for why this dataset is more accurate than any of the other cited ones? Some justification is required for using this climate dataset instead of others. In turn, more details regarding how the actual trends were computed would be helpful. Did the authors calculate the trends, or are they from prior studies? Overall, while this section provides a good overview of a particular climate dataset, it may not be robust enough to attribute glacier changes.

Page 13 line 27: Where are the weather stations located to derive this climate dataset? Were any high altitude stations used which are located nearby the glaciers?

Page 13 line 28: How were the temperature and precipitation trends derived?

Page 13 line 37: Figure 8 does not show the climate changes separated into the 1961-2000 and 2000-2013 intervals. Do you have any data suggesting that the warming rate has increased after 2000?

Figure 7: It would be nice to see the locations where each photo was taken on the map image of the glacier (in panel a).

---

## Author Comment (AC3) · 1 Aug 2018

Dear Referee #2,

Thank you for your valuable suggestions and I have already revised the article according to your suggestions. The following are a few answers to some questions.

General comments: By quantifying the rates of ice loss in the region over several decades, this study has potential to improve our understanding of multi-decade glacier changes and water resources in an important region. However, a primary concern with this paper is the large degree of overlap with a previously published work covering the

same topic in a nearby area (https://doi.org/10.5194/tc-12-103-2018). Many aspects of the methodology are nearly identical, except that a 1968 topographic map is used instead of a 1980 topographic map. I recommend the authors summarize all similar aspects, then simply cite their previous work. This will allow the new manuscript to focus on the unique portions, such as the different time intervals and the quantification of the acceleration of ice loss. This will require a significant revision by the authors. However, in its current form I feel the manuscript is too similar to the previous work for publication in TC.

Answer: Thank you for your valuable suggestions and I have already revised the article according to your suggestions. The data processing in this study was almost simultaneous with that one in Wu et al. (2018). The only difference is that the two-dimensional first-order polynomial fitting in off-glacier regions removes the residual in the differential interferogram. This study area is closed to the study area in Wu et al., 2018, and this processing scheme is similar with previous study, but the Result and Discussion have big differences, such as the distribution of debris covered region, the effect of debris-cover to mass balance, and the difference of land-and lake-terminating glacier mass balance. It is indicated that this study has great scientific value and it is definitely worth to study.

Specific comments:

(1) Page 1 line 26: The uncertainties for the 1968-2000 interval ($\pm$ 0.05) seem rather small, especially when viewing the vertical error statistics in Table 3, where the standard deviation of vertical error between TOPO and SRTM ranges from 20 to 27 meters. Figure 5a (elevation difference between 1968 and 2000) also shows large areas with significant vertical error over both ice and ice-free terrain, which may be due to interpolation procedures used when the topographic map was originally created. The limits of the color scaling (-7.69 to 7.69 meters) also seem too narrow - a wider elevation range should be used in this figure so that larger elevation changes are not saturated at the endpoints of the color bar. Based on these, I would expect uncertainties larger

than $\pm$ 0.05. I recommend careful revisit of the uncertainty estimation procedure, to ensure that the results are representative of the vertical error associated with using the historical topographic maps.

Answer: Thank you for your valuable suggestions and I have already revisited the uncertainty estimation procedure carefully and revised the result of uncertainty in manuscript and tables. For the maximum value of the color scaling, elevation differences with values exceeding $\pm$100 m were defined as outliers and omitted, so the 7.69 m a-1 is the maximum elevation difference and can be considered reasonable.

(2) Page 2 line 40: I am wondering if the authors drew the topographic maps themselves? The wording is unclear here.

Answer: The topographic maps were compiled by the Chinese Military Geodetic Service from air photos acquired in April 1968. Based on scanned and well-georeferenced topographic maps, the outlines of glaciers in the CNR in 1968 were digitized manually by authors.

(3) Page 3 line 3: Most readers will not be as familiar with the region, thus using county names may not be the best way to describe the location.

Answer: Thank you for your valuable suggestions and I have already added the latitude and longitude range of study area.

(4) Page 4 line 25: Not sure what is meant here by "higher quality images could not be acquired in 2000-2010". There are several Landsat 7 scenes obtained in the early 2000's which are cloud-free and available over the region of study (Landsat scene LE71350392001335BJC00 acquired on Dec 01 2001 for example). What is this referring to?

Answer: Thank you for your valuable suggestions and I checked Landsat 7 scenes obtained in the early 2000's in USGS, scenes with cloud-free indeed exist, but most scenes with extensive snow cover. Due to the influence of the Indian monsoon and

westerly winds, the CNR was almost permanently covered by cloud in summer and covered by snow in winter in 2000–2010, so higher quality images could not be acquired.

(5) Page 4 line 29: "No horizontal shift was observed". What is the horizontal shift being measured relative to?

Answer: I have already revised this section. "Acquired from the United States Geological Survey (USGS), the Landsat OLI images are orthorectified with the SRTM. We selected a Landsat OLI scene from 2016 as reference. For the OLI scenes no horizontal shift was observed, whereas the topographic maps had small systematic shifts of 6 to 12 m."

(6) Page 4 line 33: Do the authors have access to the original aerial photographs? If so, it would be useful and interesting to create a figure showing a sample aerial image for some of the glaciers. Also, later in in the paper, it states that Corona satellite images were used to estimate the uncertainty of glacier outlines (Page 10 line 5). Were the Corona images used in the creation of glacier outlines as well? Please clarify exactly how the aerial photographs, topographic maps, and Corona images were used to derive the glacier outlines.

Answer: I am sorry that I did not make it clear due to the mistake in the English language. I didn't have access to process the original aerial imagery but the topographic maps. Topographic maps were compiled by the Chinese Military Geodetic Service from aerial images acquired in April 1968. Based on scanned and well-georeferenced topographic maps, the outlines of glaciers in the CNR in 1968 were digitized manually by authors. For Corona image, it was used to evaluate the accuracy of glacier outlines in the western Nyainqentanglha Range (Wu et al., 2016). The image was orthorectified based on ASTER GDEM V2 and a pan-sharpened Landsat OLI image. Before digitising the glacier outlines and comparing them temporally, topographic maps had done co-registration with Corona image. After eliminating the impact of cloud and sea-
sonal snow, an accuracy of less than half a pixel was achieved. Then the offset in the pixels between topographic maps and Corona image was 2.8 m. Upon careful examination, the dark portions of the image that are included as part of the outlines, are those shadow of mountains on the glacier surface. The average offset between the topographic-map and Corona-image outlines was ±6.8 m, calculated as the average distance between points taken every 10 m along the map outline and the nearest points on the corresponding Corona image outline. Hence, average offsets between topographic-maps outlines and Corona image in CNR can be considered as the same offsets of ±6.8 m, and then mean relative error was calculated for glacier area in 1968.

(7) Page 5 lines 5-9: The logic in this statement is unclear to me. How are the ±0.8% and ±3.0% values derived?

Answer: Compared glacier outlines derived from Landsat-images with real-time kinematic differential GPS (RTK-DGPS) measurements, average offsets of ±10 m and ±30 m were acquired for the delineation of clean and debris-covered ice (Guo et al., 2015), whereas average offsets between topographic-maps outlines and Corona images was ±6.8 m (Wu et al., 2016). For glacier outlines in 1968, a buffer with ±6.8 m was generated in ArcGIS software. The difference between the area of new polygon and 1968 glacier area is the error of 1968 glacier area. Then divided the error by 1968 glacier area, mean relative error of ±0.8% was determined for glacier areas in 1968. The same method was employed for 2016 glacier area and mean relative error of ±3.0% was determined for glacier areas in 2016.

(8) Page 5 lines 25-34: A more detailed and clear description is required to understand how the glacier centerlines were derived. In what way are the glaciers divided into two polygons? Also I do not understand how and for what purpose the derived centerlines are compared with high-resolution aerial imagery. How are the uncertainties of 6 and 7.5 meters obtained? Perhaps a figure helping to illustrate this process would be helpful.

[Figure]

Answer: Based on a glacier-axis concept derived from glacier morphology that only requires glacier outlines and a digital elevation model (DEM) as input, glacier center-line was derived semi-automatically in this study (X. Yao et al., 2015). The glacier-axis concept assumes the main direction of any given glacier can be defined as a curved line. The glacier outline is divided initially into two curved lines based on its highest and its lowest elevation. Using these, the glacier polygon is then divided by Euclidean distance into two regions. The common boundary of these two regions is the glacier axis or glacier centerline. An error estimation of the resulting centerlines was performed, comparing the semi-automatically generated results to high-resolution aerial imagery at the terminus. A Corona image, with a resolution of 4 m, and Google EarthTM images, with a resolution better than 1 m, were used to evaluate the accuracy of these centerlines. In a comparison with topographic maps and Landsat images, the uncertainties in centerline location were no more than 6 m and 7.5 m, respectively.

(9) Page 8 line 9: Regarding the statement "probably overestimate the uncertainty of the larger sample". What "larger sample" is being referred to here?

Answer: "Larger sample" refers to the larger off-glacier regions. Because averaging in larger regions reduces the error, the standard deviation (SD) in off-glacier regions will probably overestimate the uncertainty of the larger off-glacier regions. Hence, the uncertainty can be estimated by the standard error of the mean (SE= SD/$\sqrt{N}$, N is the number of the included pixels).

(10) Page 8 line 13: How do the decorrelation lengths factor into the uncertainty estimates? An additional equation showing exactly how they are used would be helpful.

Answer: To minimize the effect of autocorrelation, a decorrelation length based on the spatial resolution is recommended. From previous studies, decorrelations of 600 m and 200 m were employed for different DEMs with the spatial resolution of 30 m and 10 m (Bolch et al., 2011; Paul et al., 2015). Utilization of the decorrelations of 600 m and 200 m, regular network were created in whole study area. Then the included

pixels in off-glacier area were counted and used to estimate the standard error of the mean (Table 3). The overall errors of derived surface-elevation changes can then be estimated using mean standard error and average elevation difference in off-glacier regions.

(11) Table 3: Using MED as an abbreviation for mean may be confusing, as MED is commonly used to abbreviate median.

Answer: Thank you for your valuable suggestions and I have already changed "MED" into "AED". (AED as an abbreviation of Average Elevation Difference)

(12) Page 8 Line 33: How were debris-covered portions of the glaciers delineated?

Answer: I have already introduced how to separate the debris-free and debris-covered regions in the section of 4.1. Glacier outlines in 2016 were delineated using a ratio threshold method, a division of the visible or near-infrared band and shortwave infrared band of Landsat OLI images (Paul et al., 2009; Racoviteanu et al., 2009). A 3 × 3 median filter was applied to eliminate isolated ice patches < 0.01 km2 (Bolch et al., 2010b; Wu et al., 2016). In order to discriminate proglacial lakes, seasonal snow, supraglacial boulders and debris-covered ice, scenes without snow, or cloud-free image scenes acquired at nearly the same time, were used for reference when making manual adjustments. Generated from the SRTM-C DEM automatically, topographical ridgelines (TRLs) were used to divide the final contiguous ice coverage into individual glacier polygons (Guo et al., 2015). Due to a small proportion of debris-covered regions in study area, the debris-free and debris-covered regions were separated manually using Landsat OLI images.

(13) Page 9 line 14: The magnitude of this length change seems extremely large. Is this a lake-terminating glacier? Can the authors show images of the glacier in 1968 and 2016 to confirm this? On another note, it is difficult in this paper to determine which glacier ID corresponds to which glacier in the figures. For example, the numbers in Figure 1 should correspond to the numbers in Table 6, as currently this does not

seem to be the case.

Answer: Thank you for your attention and I made a mistake in this paragraph. Glacier 5O281B0575 experienced the most recession, not Glacier 5O281B0668. And Glacier 5O281B0575 is a lake-terminating glacier. I have already added a supplementary figure S2 to show more detail of glacier length change. "Glacier 5O281B0575 experienced the most recession (105.1 m a-1), its length decreasing from 24703 m to 19657 m. The terminus elevations of these selected glaciers rose an average of 113 m, varying from 23 m (3759 to 3782 m a.s.l.) to 323 m (3981 to 4304 m a.s.l.)."

(14) Page 13 line 13: It is interesting to see the 0.5 x 0.5 degree temperature and precipitation trends derived from the Grid-based China v 2.0 dataset. However, it seems rather tenuous to base conclusions on a single climate dataset at very course spatial resolution over a region of extreme mountain topography. As the authors summarize in the paragraph starting on page 13 line 14, different climate datasets give widely varying results. The authors have chosen this particular climate dataset to base their conclusions on - is there sufficient reasoning for why this dataset is more accurate than any of the other cited ones? Some justification is required for using this climate dataset instead of others. In turn, more details regarding how the actual trends were computed would be helpful. Did the authors calculate the trends, or are they from prior studies? Overall, while this section provides a good overview of a particular climate dataset, it may not be robust enough to attribute glacier changes. Page 13 line 27: Where are the weather stations located to derive this climate dataset? Were any high altitude stations used which are located nearby the glaciers? Page 13 line 28: How were the temperature and precipitation trends derived? Page 13 line 37: Figure 8 does not show the climate changes separated into the 1961- 2000 and 2000-2013 intervals. Do you have any data suggesting that the warming rate has increased after 2000?

Answer: I have already revised this section. "To analyse the response of glaciers in the CNR to climate change, relevant air temperature and precipitation datasets were taken from the Dataset of Daily $0.5° \times 0.5°$ Grid-based

Temperature/ Precipitation in China (V2.0) (Dataset2.0). Using the thin plate smooth spline method, and a 50 year (1961 to 2010) quality controlled observational daily temperature and precipitation data series based on 2472 gauges (http://data.cma.cn/data/cdcindex/cid/00f8a0e6c590ac15.html), Dateset2.0 was produced by the Climate Data Center, National Meteorological Information Center, China meteorological Administration for Mainland China. Previous study showed that the mean bias error of precipitation in large part gauge is between -1 mm d-1 and 1 mm d-1. Dataset2.0 reduced the rain intensity when heavy rain or moderate rain comes. Over the light rain, it has more veracity. Dataset2.0 is exact describing the climate characteristic of the Tibetan Plateau, the Tienshan Mountains and Tarim Basin (Zhao and Zhu, 2015). Linear regression analysis was performed in each grid for temperature and precipitation during 1961-2010, 1961-2000 and 2000-2010. Fig. 10 shows the horizontal distribution of surface temperature and precipitation changes from May to September since 1961. It is clear that increasing surface temperatures and decreasing precipitation have been dominant in the CNR in recent decades. The changes in surface temperature and precipitation were confirmed with data from the three nearest meteorological stations, Jiali (30°40′N, 93°17′E, 4488 m a.s.l.), Linzhi (29°40′N, 94°20′E, 2992 m a.s.l.) and Bomi (29°52′N, 95°46′E, 2736 m a.s.l.). Surface temperature at these stations increased slightly from 1961 to 2000 and then significantly after 2000, the average temperatures at the three stations after 2000 increased 0.44 $\sim$ 0.50°C than those before 2000. The trend of precipitation is not evident at the three stations and present large interannual precipitation fluctuations. Dataset2.0 shows average precipitation decreasing by more than 40 mm per decade since 1961, resulting in less glacier accumulation. The reduced precipitation on the N slope is smaller than on the S slope, but glaciers on the N slope experienced a more intense mass loss than the S slope. This suggests the influence of precipitation is much less on glacier mass loss in the CNR. The average surface temperature increased by more than 0.2°C per decade in the CNR (with a confidence level <0.05), higher than the rate of global warming (0.12°C per decade, 1951–2012) (IPCC, 2013). The warming rate

on the N slope is slightly larger than that on the S slope. Furthermore, a lesser warming rate was present from 1961 to 2000, becoming greater after 2000. The changes of average surface temperature are consistent with the changes of glaciers. The mean mass deficit in the 5O28 drainage basin (on the S slope) was smaller than that in the 5N22 drainage basin (on the N slope) during the investigated periods. Glacier mass loss in the CNR can be attributed to climate warming."

This dataset was produced by the Climate Data Center, National Meteorological Information Center, China meteorological Administration and has excellent reliability. I have already added the location of weather stations in Figure 10. There have no other high altitude stations can be used which are located nearby the glaciers. The temperature and precipitation trends were derived from linear regression analysis in each grid during 1961-2010. Fig. 10 shows the horizontal distribution of surface temperature and precipitation changes from May to September since 1961.

(15) Figure 7: It would be nice to see the locations where each photo was taken on the map image of the glacier (in panel a).

Answer: Thank you for your valuable suggestions and I have already added the locations where each photo was taken on the map image of the glacier (in panel a).

Best Regards, Wu Kunpeng and other authors

Please also note the supplement to this comment:
https://www.the-cryosphere-discuss.net/tc-2018-90/tc-2018-90-AC3-supplement.zip